# MSPA-informed SLEUTH urban growth modeling for green space protection in Ottawa

Abdolrassoul Salmanmahiny[1¤*], Scott W. Mitchell[1], Joseph R. Bennett[2]

1 Department of Geography and Environmental Studies, Carleton University, Ottawa, Canada,
2 Geomatics and Landscape Ecology Research Laboratory, Carleton University, Ottawa, Canada

¤ Current address: Department of Environmental Sciences, Gorgan University of Agricultural Sciences and Natural Resources, Gorgan, Golestan, Iran
* Rassoulmahiny@gmail.com, Mahini@gau.ac.ir

## Abstract

We created optimal urban expansion scenarios that also safeguard green spaces using SLEUTH-3r in the National Capital Region, Ottawa, Ontario. The scenarios were based on using two exclusion layers in SLEUTH-3r modeling, adjustments to the model's calibrated growth coefficients for a compact city scenario and applying green space social equity weights to urban zones in model's prediction results. The first exclusion layer contained common restricted areas for urban growth, while the second additionally incorporated cores of green spaces defined through Morphological Spatial Pattern Analysis (MSPA), core importance and their corridors for connectivity. For each scenario, we selected 23,850 hectares as the required urban growth by the year 2050 and only 10% of this amount (2385 ha), to encourage more compact growth. We compared the scenarios based on the affected green space cores and urban growth polygons using Technique for Order of Preference by Similarity to Ideal Solution (TOPSIS). In most cases, scenarios incorporating MSPA were the favored ones. As the first attempt integrating MSPA definition of green space cores, their importance and connectivity into SLEUTH-3r model, we showed that MSPA-informed SLEUTH-3r modeling affects prediction results and provides a useful platform for generating scenarios. Incorporating MSPA information into SLEUTH-3r modeling enhanced the protection of green space cores and their connectivity. However, it also led to the selection of smaller urbanization polygons for the year 2050, distributed across the study area. Focusing on the preferred options, social equity weights and the selected polygons, provides city planners and stakeholders with valuable assistance and flexibility in designing urban growth scenarios while protecting green spaces.

**Data availability statement:** The address to access the data through Zenodo is mentioned at the end of the text. The address is: https://zenodo.org/records/14752824

**Funding:** Part of the funding for the research presented in this paper was provided for the

sabbatical studies of the first author by Gorgan University of Agricultural Sciences and Natural Resources, Gorgan, Iran. Additional funding was kindly provided by Carleton International and by the Department of Geography and Environmental Studies at Carleton University. There was no additional external funding received for the study. The funders had no role in study design, data collection and analysis, decision to publish or preparation of the manuscript.

**Competing interests:** The authors have declared that no competing interests exist.

## 1. Introduction

Currently, over half the world's population resides in urban areas, and this is expected to rise to nearly 70% by 2050 [1,2]. This rapid urbanization has led to challenges such as biodiversity loss and habitat fragmentation and destruction [3–9]. Improved urban expansion planning can help mitigate these impacts. Predictive tools like the cellular automata–Markov (CA-Markov) model support such planning by forecasting changes in urban and other land use/land cover (LULC) types [10]. Cellular automata are discrete spatiotemporal systems governed by local rules [11], typically represented on raster grids derived from remote sensing data.

SLEUTH is a CA-Markov urban growth modeling tool [12,13] and requires one slope layer in percent, 2 LULC layers at different times, an exclusion layer depicting areas where development is prohibited or impossible, 4 urban layers representing previous urbanized areas, at least 2 transportation layers depicting roads at different times in the past, and a hillshade layer (utilized for visualization purposes only). Different exclusion layers can affect SLEUTH calibration and prediction [14–17]. SLEUTH employs 4 growth rules: Spontaneous Growth, New Spreading Center, Edge Growth, and Road-Influenced Growth. These 4 growth rules are governed by 5 urban growth coefficients: Diffusion or dispersion, Breed, Spread, Slope Resistance, and Road Gravity. For further information on the growth rules and coefficients, readers are referred to [12] and [18].

Using SLEUTH involves derivation of the 5 urban growth coefficients through three calibration steps: coarse, fine, and final. Dietzel and Clarke [19] proposed the Optimal SLEUTH Metric (OSM), comprised of fit metrics computed by SLEUTH, as best for modeling success assessment. After calibration, utilizing the refined coefficients, the two most important outputs generated for the target year in the future are an *urbanization likelihood layer* ("cumulate_urban") and a *predicted LULC* layer. The former assists in identifying the image pixels most probable to transition into urban areas up to the target year while the latter shows LULC in the target year.

SLEUTH has proven effective for managing water quality in the Chesapeake Bay [18], guiding Beijing's future expansion [20], assessing land use policies [21], and enhancing planning accuracy through informed exclusion layers [15,17,22]. Its enhanced version, SLEUTH-3r, offers improved computational efficiency and flexibility, allowing modification of parameters like the Diffusion Multiplier (DM) for more accurate predictions [14,23]. SLEUTH-3r has been applied in studies across Baltimore, USA [24], Groningen, Netherlands [25], and Ningxia and Shizuishan, China [23,26]. However, optimal input image resolution remains uncertain, and the role of integrating new data layers into the exclusion layer is still not fully explored. This study offers insights into broadening the potential applications of the SLEUTH-3r model.

One approach to manage the effects of urbanization on green spaces entails generating scenarios of urban growth while considering their impacts on green space cores, core importance and connectivity. Cores can be defined using Morphological Spatial Pattern Analysis (MSPA) for ecology [27–29]. MSPA distinguishes categories such as core, edge, perforation, islet, bridge, loop, and branch [28,29]. Core importance and connectivity can be calculated using tools such as Conefor [30]

and Circuitscape software [31], respectively. Including this information into the exclusion layer of SLEUTH-3r offers an approach to expand the existing application scope of this model.

MSPA has been used for effective green infrastructure and biodiversity conservation planning, while considering urban growth [31–34], and core importance and Conefor software application have been the major themes of relevant recent studies [35–37]. Some of the recent studies using Circuitscape software include: modeling connectivity in heterogeneous landscapes for conservation [29], considering diverse land uses and stakeholders' interests [38,39], sustainable urban development [40], identifying urban ecological security patterns [41], finding key sites for forest habitat connectivity restoration [42], and protecting rare species [43].

As an innovation, green space core assessment is integrated into SLEUTH-3r modeling using MSPA, Conefor, and Circuitscape. This allows for the generation of urban growth scenarios influenced by green space features. These scenarios can be evaluated using Multi-Attribute Decision Making (MADM) methods. One common method is the Technique for Order of Preference by Similarity to Ideal Solution (TOPSIS) [44]. It helps identify the best option when multiple alternatives with different weights are present. There is a research gap on how the inclusion of green space cores affects SLEUTH-3r results. It is also unclear how this approach can support scenario generation and selection for Ottawa. This is especially relevant due to Ottawa's rapid urban growth and the resulting pressure on green spaces. The study assumes that green space protection can be achieved through scenario optimization. It also proposes that such strategies are applicable to other rapidly urbanizing areas like the study region.

Given the insights offered in the above discussion, we selected SLEUTH-3r to model urban growth within the National Capital Region of Canada. Our goal was to develop and select the preferred scenarios for urban expansion while protecting green spaces using SLEUTH-3r. In using SLEUTH-3r, we uniquely focused on green space cores, their importance and connectivity. We evaluated future urban growth scenarios using TOPSIS to identify the best areas for urbanization, enhance protection of green spaces, promote compact city growth where possible and ensure equitable distribution of green space across urban zones.

## 2. Materials and methods

### 2.1. Study area

Canada's Capital Region (NCR) holds official federal designation as the Canadian capital encompassing Ottawa, Ontario, and the neighbouring city of Gatineau, Québec, along with surrounding suburban and exurban communities [45]. According to the National Capital Act (1985), the National Capital Region spans an area of 4,715 km$^2$ situated along the Ottawa River, which serves as the boundary between the provinces of Ontario and Québec (Fig 1). With a population of 1,488,307 as of 2021 [46], this extensive and predominantly flat expanse is primarily covered by urban areas, green spaces, agriculture, protected areas, and parks. We selected an enclosing rectangular area slightly larger (810000 hectares) than the NCR to represent the region (Fig 1). This area has recently undergone rapid urban growth with nearly 29,885 hectares of urban growth between 1990 and 2020, threatening green spaces and justifying the need to focus on future city growth projections and effective management strategies.

### 2.2. Data source

We used the Canadian Digital Elevation Model (CDEM) [47] to generate slope and hillshade layers in QGIS v3.28 [48]. For land cover, GLC_CFS30 [49] was the primary source, supplemented by urban area data from Dynamic World [50] and GAIA [51]. Common urban areas across these datasets were merged and refined through visual inspection. We selected 1990 and 2020 as the start and end years for SLEUTH-3r modeling, ensuring sufficient temporal coverage and data availability. High-resolution geo-registered Google Earth images from 1990, 2000, 2010, and 2020 supported independent accuracy assessments. To streamline modeling, GLC_CFS30 classes were reclassified into six categories: agriculture, forest/vegetation, wetlands, urban, bare areas, and water bodies. Using 90 random samples per class, overall accuracy was 94%, which was suitable for our modeling needs.

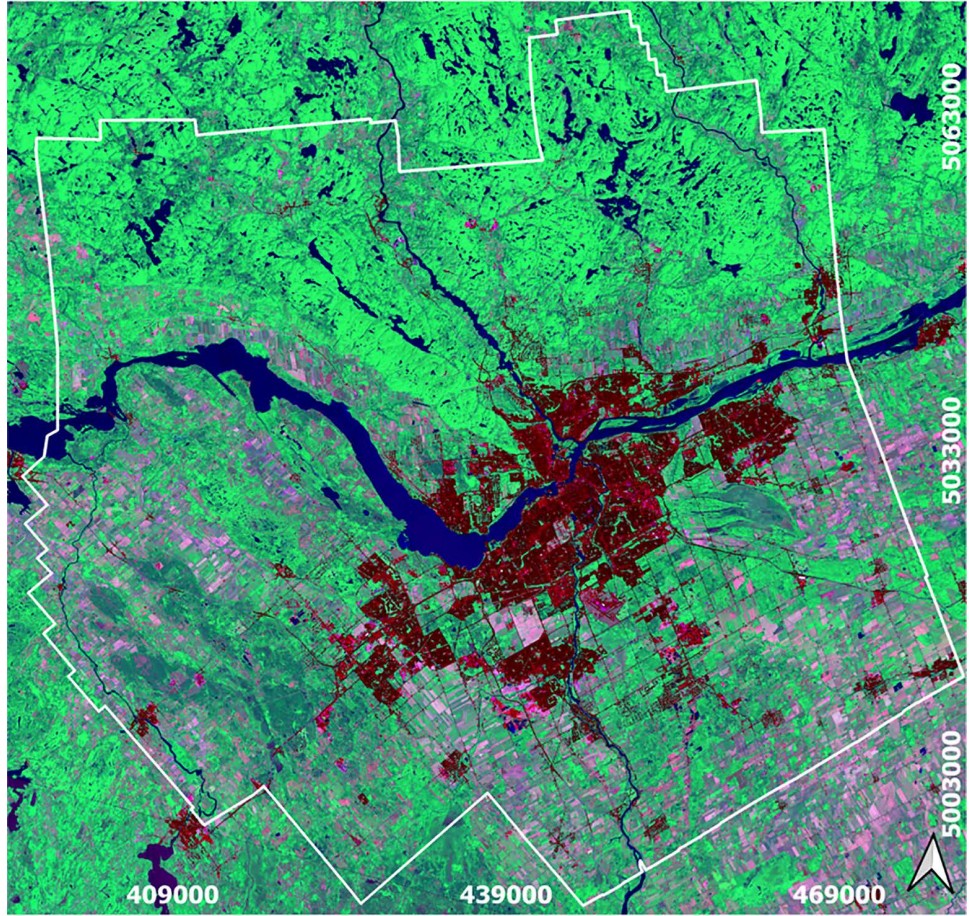

**Fig 1. Boundary of the national capital region and urban areas (dark red) on a Landsat color composite map (USGS, 2020) of the study area.**

To protect the green spaces, we prepared two versions of the exclusion layer as a basis for our first scenario: Exclusion 1 included waterbodies, roads, crown land use planning, ecological corridors and protected areas, Gatineau Park, city green belt and green spaces, floodplain and inundation zone for Ottawa River, public and urban land designations, and zone subtypes. Exclusion 2 incorporated additional information obtained for the green spaces, including forests, orchards, woodlands, and green landscapes. For this, we applied MSPA classification of the green spaces, then selected green space cores—including cores, perforations, and edges [52]—and submitted the result to *core importance* evaluation in Conefor software [28]. For *core importance* we used the area of the cores and their distances from each other in Conefor. The table values resulted from this application were then applied to the green space cores, where higher values indicated greater importance of the cores for protection.

Next, we assessed corridors based on a *current map* generated using Circuitscape software [29]. For Circuitscape, we used the focal points of the green space cores and a conductance layer that was generated through assigning higher values in land use map to forest patches and progressively lower values to wetlands, water bodies, agricultural areas, roads, urban areas, and bare areas. *Current maps* help identify critical pathways for movement and connection between habitat patches. High current areas indicate regions where connection is easier, while low current areas may act as barriers. Focusing on high-current regions ensures better connectivity among the selected cores. We then combined the *core importance* layer and *current map* layers, then overlaid the result on the original exclusion layer to create Exclusion

2 layer. In this way, Exclusion 2 included additional information on the core importance and the connection of green space cores, preventing SLEUTH-3r from attempting to urbanize areas with high green space core importance or connectivity. The two exclusion layers were used in distinct calibration and prediction processes.

Transportation data were obtained from OpenStreetMap [53]. The 2020 road network was used as a reference to visually identify and remove roads not present in 1990, 2000, and 2010, using geo-registered Google Earth images. To ensure coverage, all major roads were included through a full-area visual review. Minor roads missed in earlier years were assumed to have minimal impact on SLEUTH-3r results. All raster layers were standardized to 1000 × 1000 cells at 90 × 90 m resolution and projected to WGS84 UTM Zone 18N. Table 1 summarizes the input data sources used in this analysis.

## 2.3 Methods

### 2.3.1. Core importance and corridor current map.

For *core importance* we calculated the probability of connectivity (PC) and the importance value of green space patches (dPC) in terms of their contribution to overall green space availability and connectivity [28] using formula (1) and (2) as below:

$$PC = \frac{\sum_{i=1}^{n} \sum_{j=1|}^{n} a_i \cdot a_j \cdot p_{ij}^*}{A_L^2} \tag{1}$$

$$dPC = \frac{PC - PC_{remove}}{PC} \times 100\% \tag{2}$$

where $n$ represents the total number of green space cores in the region; $ai$ and $aj$ are the areas of green space core $i$ and $j$, respectively; $p_{ij}^*$ is the maximum product of the probability of all paths between green space core $i$ and core $j$; $A_L$ is the total area of the study area. In the above formulae the greater the PC value is, the higher the connection degree of the core will be. Also, in formula (2) dPC indicates the importance of any given green space core; and $PC_{remove}$ shows the probability of the connectivity after the removal of this core [54]. For this calculation, we used green space core sizes and their distances to each other as two input files for the Conefor software. The result of dPC was then assigned to the green space cores for further processing [55].

For corridor assessment, we assigned conductance values to the initial LULC types, overlaid with increasingly higher-valued islets, branches, loops, and bridges detected using the MSPA analysis of the 2020 LULC layer. Using this layer as a conductance map and the center point of each green space core as focal nodes, we generated a *current map* in Circuitscape software. This map highlighted important areas for corridor selection to connect green space cores. The results from applying Conefor for *core importance* (Fig 2) and using Circuitscape for the corridor *current map* are displayed in Fig 3.

**Table 1. Data sources used in this study.**

| Data Layer | Data Source |
|---|---|
| Slope | CDEM |
| Land Use | GLC_CFS30, Dynamic World and GAIA, on-screen digitized roads, OSM roads and Google Earth images |
| Exclusion | The edited land use maps, and layers of parks and protected areas, ecological land masses, crownlands, greenspaces, green belts, flood prone areas, public lands, and urban land designations. |
| Urban Areas | The edited LULC maps |
| Transportation | OSM, on-screen digitization on Google Earth images |
| Hillshade | CDEM |

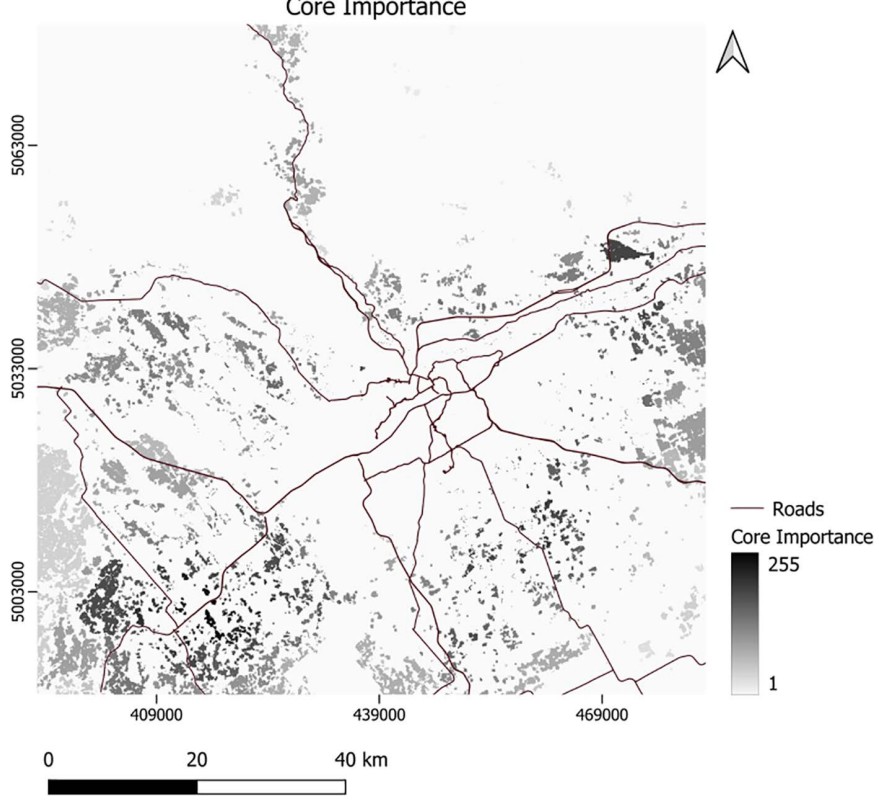

**Fig 2. Core importance Higher values indicate more important areas.**

**2.3.2. Input layers to SLEUTH-3r.** Of the required inputs to calibrate SLEUTH-3r to the study area, below we only show the two versions of the exclusion layers (Figs 4–5). In these exclusion layers, the higher values indicate more resistance to urbanization. To save space, we display other layers including slope, land use, urban areas, roads and hillshade in the supporting information section (S1–S12 Figs).

**2.3.3. Running SLEUTH-3r.** The selection of optimal $D_M$ is supposed to effectively counter SLEUTH's inclination towards edge growth. To determine the optimal $D_M$, we followed the method presented by Jantz et al. [14]. For this, the diffusion coefficient in SLEUTH-3r scenario file was initially set at 100, while all other coefficients were set to 1. Forty-two calibration processes each consisting of 25 Monte Carlo iterations were conducted, beginning with a $D_M$ of 0.001 and incrementing by 0.003 until reaching a value of 0.124. The resulting ratio files were evaluated for cluster fractional difference (CFD) and choosing the best $D_M$.

We completed the coarse, fine, and final calibration steps of SLEUTH-3r using 6, 8, and 10 Monte Carlo iterations, respectively. After each step, the Optimal SLEUTH Metric (OSM) was calculated. This metric is the product of several fit metrics generated by SLEUTH, including Compare, Pop, Edges, Clusters, Slope, Xmean, Ymean, and F-Match. After each step, the top three highest-ranking rows were then selected, and the ranges of the initial coefficients—Diffusion, Breed, Spread, Slope Resistance, and Road Gravity—were progressively narrowed. The initial range for these coefficients is 0–100, and through the calibration we determined the final values specific to urban growth in our study area. For further information readers are referred to [14,19].

We initially tested $D_M$ determination and SLEUTH-3r modeling results using 30, 60 and 90 m pixel sizes. SLEUTH-3r generated acceptable results for our study area using 1000 by 1000 grids with 90 by 90 m pixel size. For results

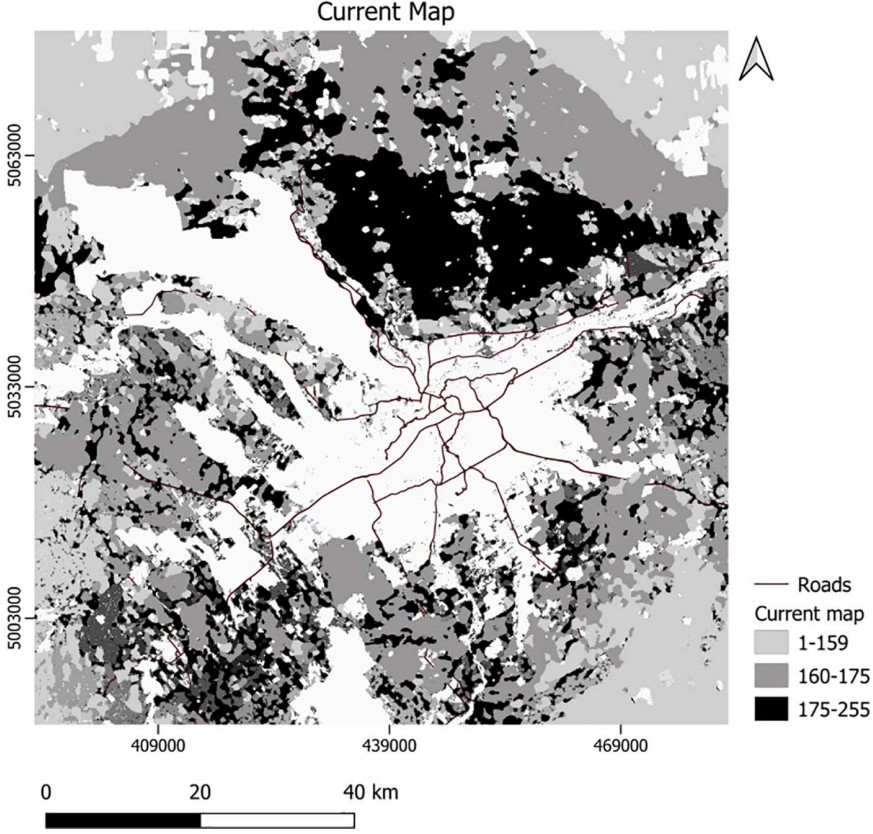

**Fig 3. Current map or corridor importance.** Higher values indicate more conducive areas.

acceptability we looked at the final *urbanization likelihood layer* generated through the modeling, and the accuracy metrics mentioned below. We used Exclusion 1 and Exclusion 2 layers in our SLEUTH-3r modeling and prediction and compared the results.

**2.3.4. Accuracy assessment of results.** The results from the coarse, fine and final calibration steps underwent evaluation using OSM as the most commonly applied method [19]. In the three steps of SLEUTH calibration—coarse, fine, and final—the OSM was calculated in Excel spreadsheet. The top three ranking rows based on OSM were selected. As mentioned above, progressively narrower parameter ranges for Diffusion, Breed, Spread, Slope resistance, and Road gravity were chosen for the next calibration step.

**2.3.5. Validation of results.** Upon finalizing the coefficients, predictive tests were conducted, setting the start year as 1990 and the end year as 2020. The modeling success was assessed using the *urbanization likelihood layer* of the year 2020 generated through prediction mode and the binary urban layer of the year 2020. These two raster layers were subjected to Receiver Operating Characteristic (ROC) and Precision-Recall (PR) metric calculations. The inclusion of the PR metric was deemed useful due to the imbalanced ratio of urban to non-urban areas in the study region. We also compared the predicted LULC for 2020 with the prepared LULC layers of the years 2020 and 1990 using a Figure of Merit (FoM) accuracy test because it directly evaluates the model's ability to predict change, whereas the Kappa Index is more suitable for overall classification accuracy but may underperform in dynamic change scenarios. Finally, the modeling results were compared visually using the actual 2020 LULC image. The finalized coefficients were utilized to forecast probable urban and non-urban changes until the year 2050.

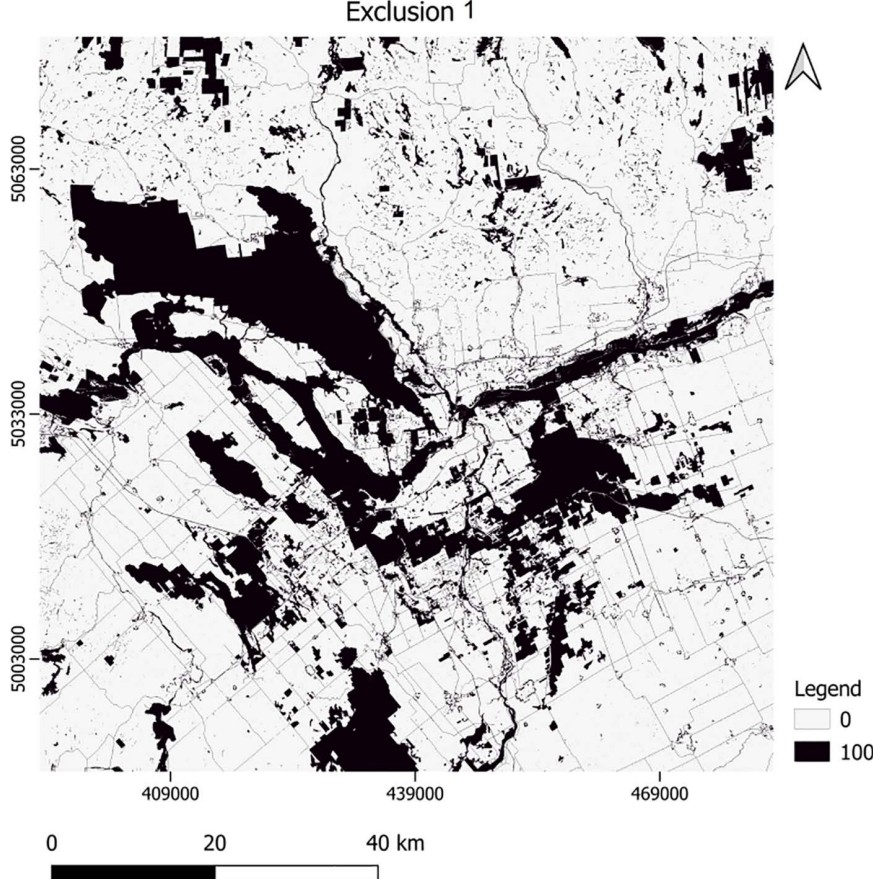

**Fig 4. The first exclusion layer used for the modeling.**

**2.3.6. Urban growth prediction.** To identify pixels with the highest likelihood of transitioning to urban areas, we used the *urbanization likelihood layers* of the year 2050 from the prediction step. As a rough estimate of the necessary urban area by the year 2050, we correlated urban area in the years 1990, 2000, 2010, and 2020 and Ottawa's population in those years. Using the projected Ottawa's population until 2050 based on the lowest growth rate as a scenario of population change, we approximated the corresponding urban area requirement for this scenario.

**2.3.7. Compact city scenarios.** In SLEUTH-3r modeling, a common method to generate scenarios is to modify the derived growth coefficients [56–58]. Here, we generated compact city scenarios by lowering Diffusion, Breed, Road Gravity, and increasing Slope Resistance coefficients, all by a ratio of 50%. A lower diffusion value results in less scattered urban development, while a lower breed value reduces the likelihood of new urban centers forming from existing developed areas. Lower spread values slow the outward growth or "sprawl" of urban centers. A lower road gravity value indicates less attraction for urban expansion along transportation corridors, whereas higher slope resistance values discourage development in areas with steep elevation changes.

This process was iterated several times to arrive at the optimal coefficients, promoting compact urban growth. In this way, urban growth experiences less spread, with reduced influence from roads and increased resistance in sloped areas. As a result, a more compact urban form is achieved, promoting multi-story buildings and high-density developments. As another form of the compact city scenario, we envisioned a future urban growth of only 10 percent of the estimated growth

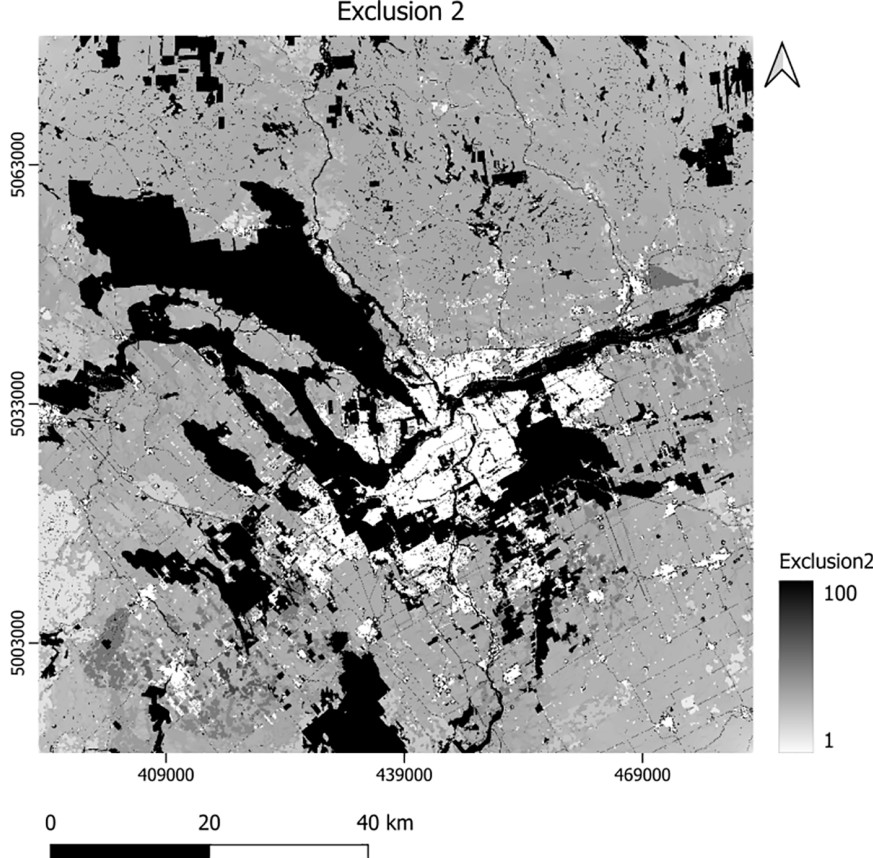

**Fig 5. The second exclusion layer additionally including core and corridor importance information.**

required by the year 2050, encouraging the development of multi-story high-rise buildings. Whether these scenarios lead to economic imbalances should be explored in future studies.

**2.3.8. Social equity in green spaces scenario.** For social equity in green spaces, we divided the study area into four cardinal direction zones. For each zone, we calculated the urban area and green spaces within 1 Km of the urban areas in the year 2020, and allocated the population to each cardinal direction [59]. We then calculated the ratio of green spaces to urban areas and to population in each direction. Using these ratios, we defined initial weights to be assigned to the *urbanization likelihood layer* of SLEUTH-3r, promoting more equitable green space distribution in each zone. The weights can be adjusted according to other factors as well and the expert recommendations in future applications of the model to evaluate their impact on the final outcomes and avoid likely social imbalances.

**2.3.9. Comparison of results.** Based on the two exclusion layers, modified growth coefficients and application of weights to the image zones, we constructed eight most likely scenarios (Table 2). To compare the scenarios, we selected groups of pixels or polygons that satisfied the minimum rank (suitability), minimum area, and total area thresholds on the *urbanization likelihood layer* generated by SLEUTH-3r. We based our comparison on visual inspection and the importance of the affected green space corridors and cores and the affected core area. We also employed the mean perimeter-to-area (PARA_MN) and mean Euclidean nearest neighbour distance (ENN_MN) metrics of the affected cores and the selected urbanization polygons using Fragstats [60]. Using TOPSIS, the scenario with the least impact on green

**Table 2. Scenarios used in this study.**

| No | Scenario | Description |
|---|---|---|
| 1 | **Usual Growth** | Exclusion 1, Current Coefficients, Un-weighted Urbanization Likelihood Layer |
| 2 | **Compact Growth** | Exclusion 1, Modified Coefficients, Un-Weighted Urbanization Likelihood Layer |
| 3 | **Social Equity Growth** | Exclusion 1, Current Coefficients, Weighted Urbanization Likelihood Layer |
| 4 | **Compact-Social Equity Growth** | Exclusion 1, Modified Coefficients, Weighted Urbanization Likelihood Layer |
| 5 | **MSPA-Informed Growth** | Exclusion 2, Current Coefficients, Un-weighted Urbanization Likelihood Layer |
| 6 | **MSPA-Informed Compact Growth** | Exclusion 2, Modified Coefficients, Un-weighted Urbanization Likelihood Layer |
| 7 | **MSPA-Informed Social Equity Growth** | Exclusion 2, Current Coefficients, Weighted Urbanization Likelihood Layer |
| 8 | **MSPA-Informed Compact Social Equity Growth** | Exclusion 2, Modified Coefficients, Weighted Urbanization Likelihood Layer |

spaces cores, their connectivity, and the best performance in terms of the selected Fragstats metrics was identified as the preferred one. The flowchart of the study is shown in Fig 6.

## 3. Results

Based on the reclassified LULC layers mentioned above from 1990 to 2020, we observed expansion of urban areas from 4.25% to 7.94%, representing approximately 29,885 hectares of urban growth. This growth is scattered across the landscape. Over the same period, agricultural lands, forests, wetlands, and waterbodies experienced a reduction in size. Roads mainly expanded during the period 1990–2010.

Within our study area, employing 1000 by 1000 raster layers with 90 by 90 m pixel size and a $D_M$ value of around 0.005 was found acceptable to simulate this growth pattern. The results of SLEUTH-3r calibration on two exclusion layers are presented in Table 3. The halved values for Diffusion, Breed and Road Gravity, and increased Slope Resistance as compact city scenarios are also included in Table 3 (bold values). Also included in Table 3 are the results of prediction accuracy assessment using the *urbanization likelihood layer* and modeled LULC of the year 2020.

In Table 3, urban growth under Exclusion 1 shows low Diffusion, moderate Breeding, and less-than-moderate Spread, while encountering relatively high Slope Resistance and a minimal influence of roads. In Exclusion 2, Diffusion and Road Gravity are higher, with less Slope Resistance and Spread, while Breeding remains nearly the same. For the compact scenarios (bold values), Diffusion, Breeding, Spread, and Road Gravity are halved, while Slope Resistance is increased by 50%. Accuracy metrics are not calculated for the compact scenario.

Using the lowest population growth rate of 1.08% observed for the year 2020, and a projection for Ottawa from 1951 to 2035 [61], we found that by 2050 Ottawa's population will peak at around 1,922,685, a potential growth of approximately 530,000 people. By analyzing images from 1990, 2000, 2010, and 2020 and examining the relationship between built and infrastructure area and population size, we estimated that around 450 square meters is directly and indirectly used per capita. Given the stability of this urban area requirement, the city will require 23,850 hectares of built and infrastructure area. Applying the two exclusion layers and $D_M = 0.05$, SLEUTH-3r generated *urbanization likelihood* and *predicted LULC* layers that met the urban area requirements for 2050.

An assessment of the ratio of green spaces to urban areas and to population for the year 2020 revealed better conditions in the north and west zones of the study area. The south zone followed, while the east zone was in the worst condition. Consequently, we assigned weights of 0.4, 0.3, 0.2, and 0.1 to the north, west, south, and east zones, respectively. These weights were applied to the *urbanization likelihood layer* when selecting suitable polygons for urbanization in the year 2050.

We developed Python code to select the top 10 polygons for each scenario (Fig 7), setting the minimum suitability at 50 and the minimum polygon size at 30 hectares. Consequently, the code selected 756 hectares for the usual scenario, 761 hectares for the compact scenario, 504 hectares for the MSPA-Informed scenario, and 520 hectares for the

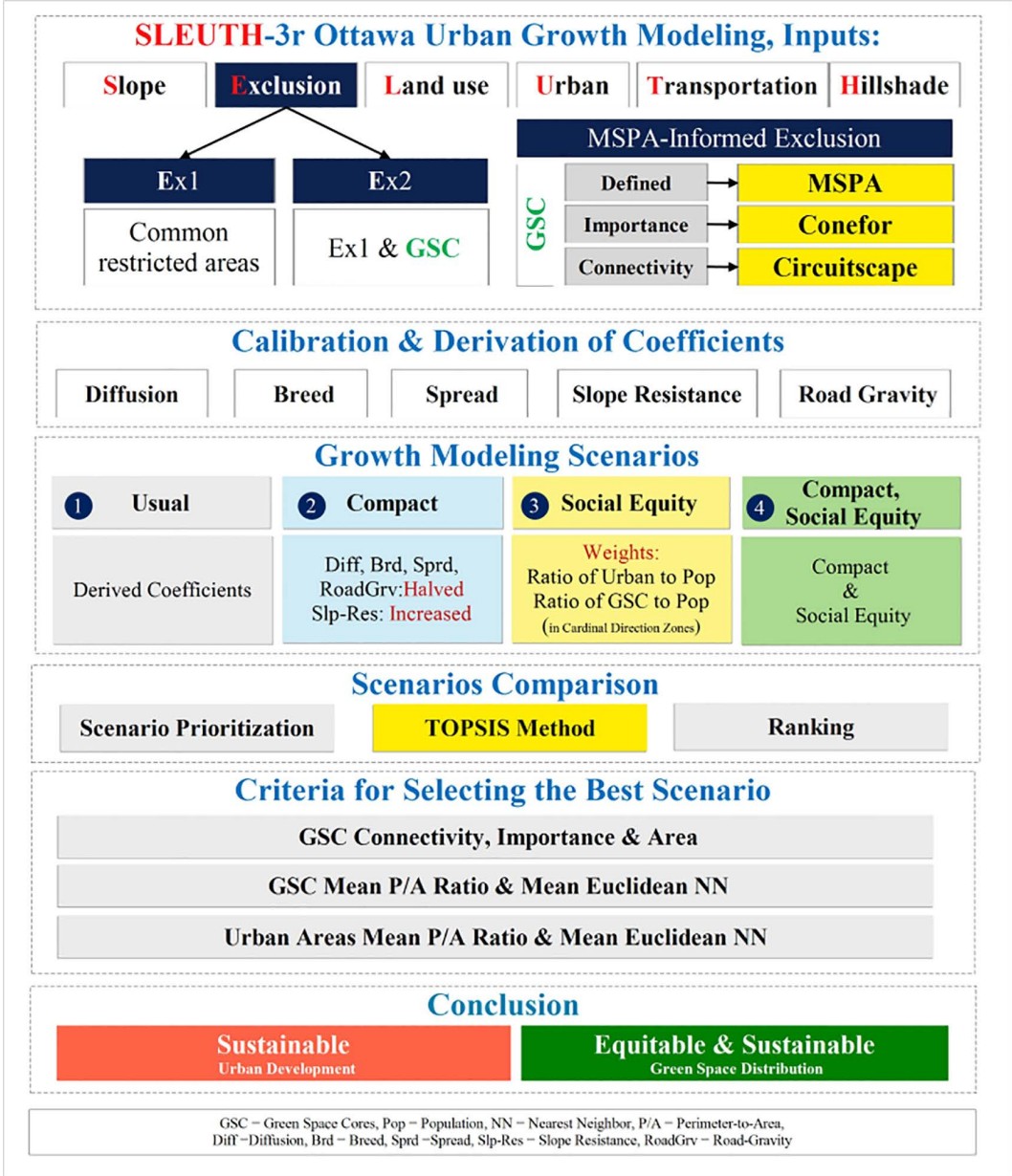

**Fig 6. Flowchart of the study.**

**Table 3. SLEUTH-3r calibration and modeling accuracy assessment.**

| Exclusion Layers | Calibration Results | | | | | Modeling Accuracy | | | |
|---|---|---|---|---|---|---|---|---|---|
| | Diffusion | Breed | Spread | Slope Resistance | Road Gravity | OSM | ROC | Precision-Recall | FoM |
| Exclusion 1 | 6 | 45 | 30 | 60 | 12 | 0.52 | 0.94 | 0.85 | 0.88 |
| Exclusion 1, Compact | **3** | **22** | **30** | **90** | **6** | | | | |
| Exclusion 2 | 29 | 47 | 20 | 24 | 30 | 0.54 | 0.85 | 0.72 | 0.94 |
| Exclusion 2, Compact | **15** | **23** | **20** | **36** | **15** | | | | |

MSPA-Informed compact scenario (Fig 8). Applying weights for the social equity scenario resulted in different patterns of the top 10 selected polygons for urbanization, prioritizing the northern portion of the area (Fig 8). We also experimented with selection of a maximum 23850 ha (the area needed to support the projected growth) of the best ranking urbanization polygons (Figs 9–10). To enforce more compact city growth, we selected 2385 ha of the highest-ranking pixels, equal to 10 percent of the estimated required land needed to accommodate growth by the year 2050 (Figs 11–12).

In these figuresurban growth patterns under varying scenarios are illustrated using three polygon styles for clarity:

- Hollow polygons: areas selected under the Compact (left) or MSPA-Informed Compact (right) scenarios, both without social equity weighting.

- Solid polygons: areas selected when the exclusion layer includes social equity (left) or social equity plus MSPA connectivity (right).

- Horizontally hatched polygons: overlapping areas selected in both left and right scenarios, indicating agreement.

In Fig 7, (Compact without MSPA) shows larger and more contiguous urban growth, mainly along a northeast–southwest axis. Fig 8 (MSPA-Informed Compact with social equity) shows smaller, more constrained growth, with reduced overlap, reflecting ecological priorities. Figs 9 and 10 follow a similar pattern. Fig 9 (Compact plus

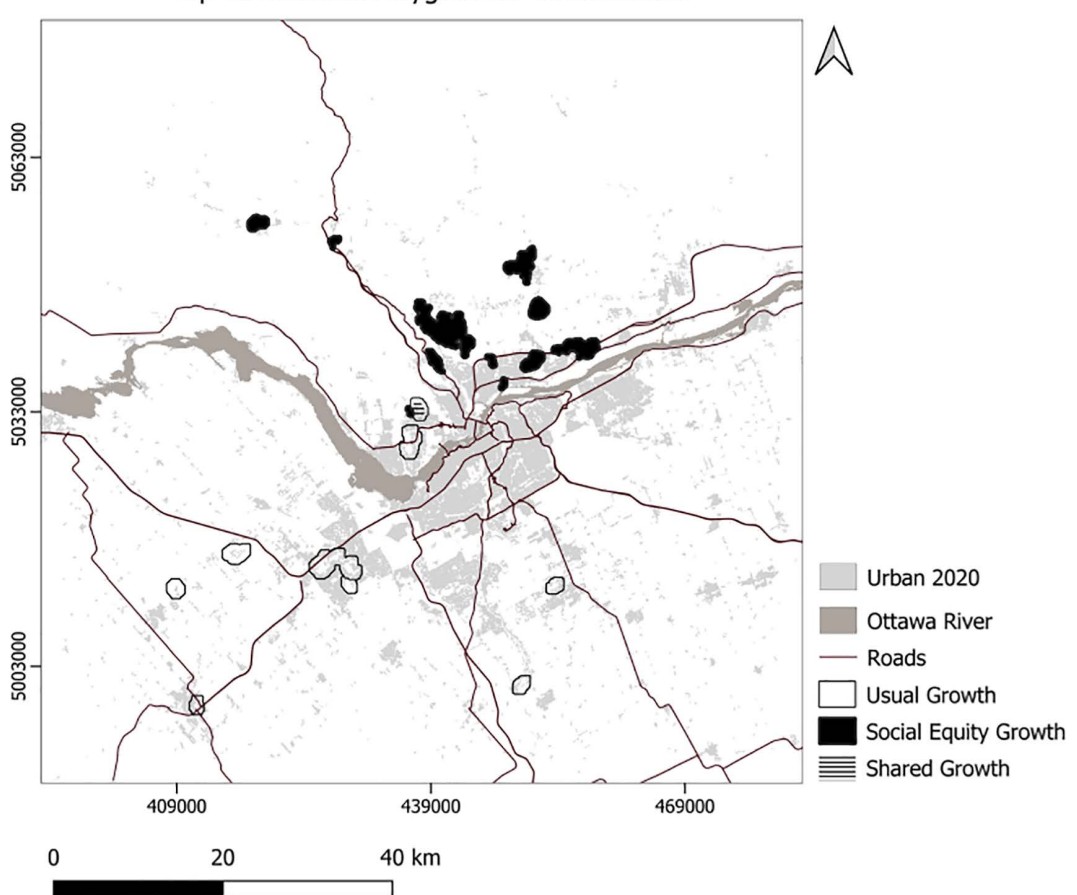

**Fig 7. Generalized top 10 selected polygons for urbanization using usual and social equity scenarios.**

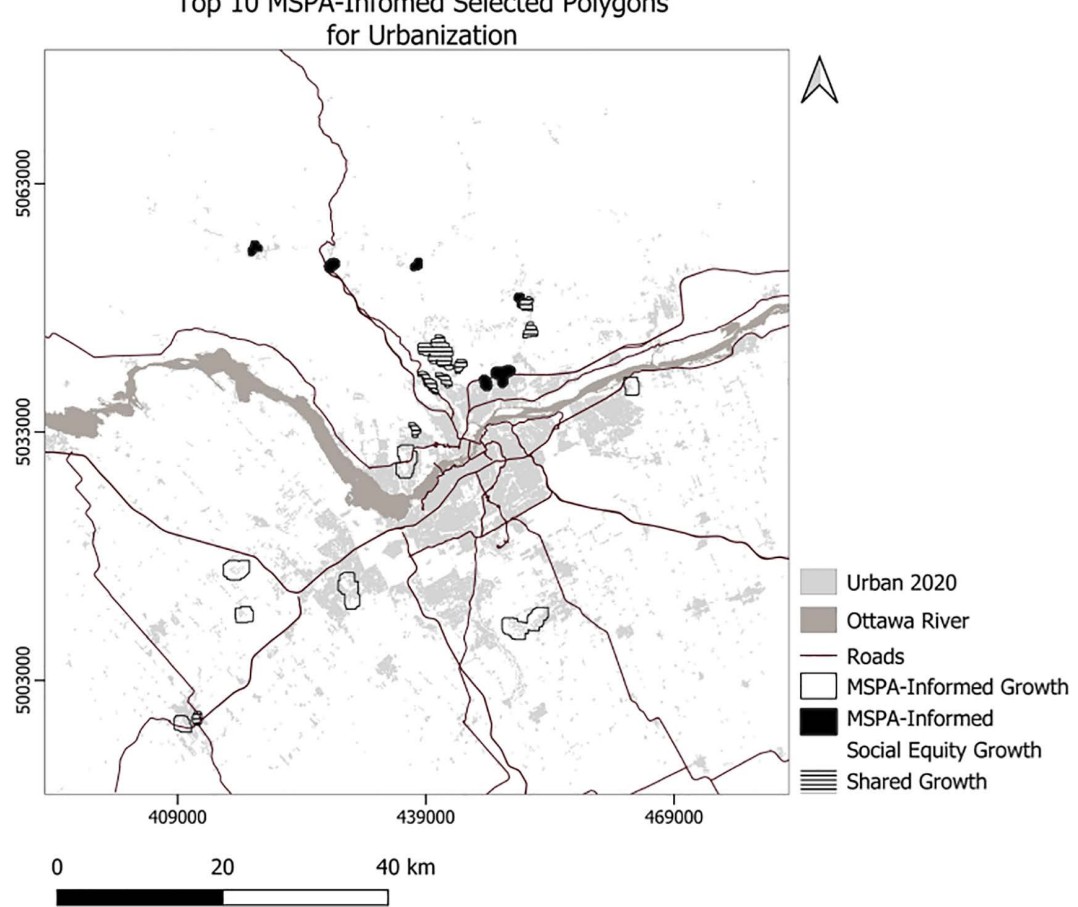

**Fig 8. Generalized top 10 selected polygons for urbanization using MPSA-Informed and social equity growth.**

Social Equity) shows more centralized growth, while Fig 10 (MSPA-Informed plus Social Equity) has smaller, dispersed patches, though shared areas appear in the southeast. In Fig 11, (Compact plus Social Equity) and Fig 12 (MSPA-Informed plus Social Equity) maps both indicate a shift of growth to the northeast when MSPA is used. Urban patches are generalized using a mode filter for clearer visualization. Under the 2,385-ha scenario, the northern region is preferred.

Overall, spatial differences across the left and right maps in each figure reflect how social equity and ecological connectivity influence the location, shape, and extent of urban expansion. These comparisons help planners evaluate trade-offs between competing priorities.

The effects of selecting 23,850 hectares for urban growth on the corridor (current map) and core importance maps were assessed in percentage using threshold above 150 (in the range of 0–255) and the area of green space cores affected by each scenario. The same assessment was applied to the best 2,385 hectares of the selected areas (Fig 13).

Table 4 lists the input data used for scenario selection via the TOPSIS method. To emphasize green space cores, the highest weight was given to the percent of affected core area, followed by high-quality cores (>150) and corridors, based on expert judgment. Fragstats metrics for affected core areas were also weighted more heavily than those for urbanized areas. To ensure robustness, we performed 20,000 iterations of TOPSIS rankings, allowing up to 50% variation in initial weights. In each iteration, a small portion of one factor's weight was shifted to another, and TOPSIS

**Fig 9. The generalized selected urbanization polygons using usual and social equity scenarios.**

was recalculated. This process continued until the 50% threshold was reached. Due to the small number of factors, this approach made using other methods like AHP or entropy unnecessary. The iterative results closely matched the initial rankings shown in Figs 14–15.

As demonstrated in Fig 14, for 23,850 ha of new urbanization areas by the year 2050, the best scenario in terms of the affected high-quality corridors is "Compact-Social Equity", whereas the best scenario in terms of the affected high-quality cores is "MSPA-Informed Social-Equity". However, the best scenario in terms of the affected core area is "MSPA-Informed Compact-Social Equity". In Fig 15., for 10 percent of urbanization (2385 ha), the best scenario in terms of the affected high-quality corridors is "Usual Growth" whereas the best scenario in terms of the affected high-quality cores and core area is "MSPA-Informed Compact-Social Equity".

Scenario rankings vary with weight choices. Using PARA_MN and ENN_MN for cores and urban areas, the top scenarios for 23,850 ha and 2,385 ha urbanization by 2050 are "MSPA-Informed Compact-Social Equity" and "Compact-Social Equity," respectively. The latter is closely followed by "MSPA-Informed Social Equity," highlighting the value of MSPA-Connectivity data. Assigning lower weights to urbanized areas in Table 4 tends to favor MSPA-Informed scenarios, if planners are willing to accept less compact but slightly more irregular polygons of urban growth forms to better protect green spaces.

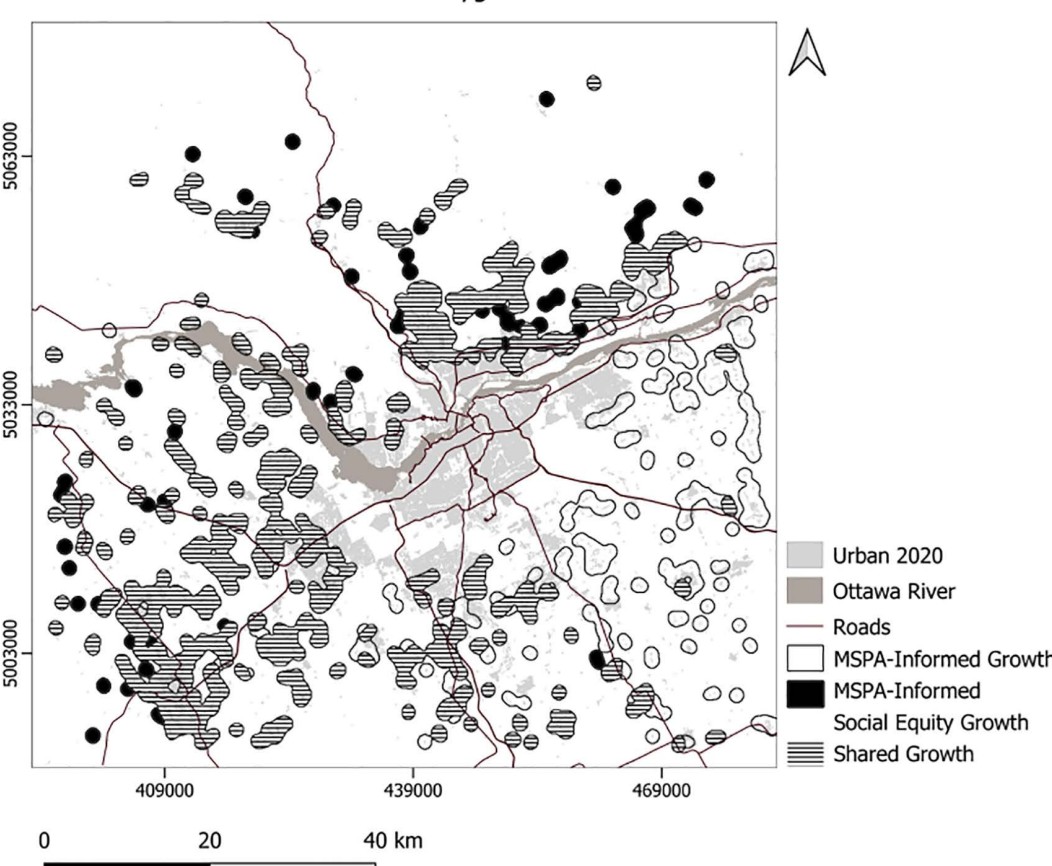

**Fig 10. MSPA-Informed, and social equity growth.**

## 4. Discussion

Interest in managing green spaces in urban areas and their peripheries, as well as the effects of green spaces on urban quality, has grown significantly (e.g., [62–65]). A key trend in this context focuses on the morphological spatial pattern analysis (MSPA) of green spaces and their relationships with urban growth. Examples include green infrastructure connectivity in Germany's Ruhr area [66], and in Beijing, Taihu Lake, Chengdu, Fuzhou, and Tongxiang cities, China [67–71]. As an expansion of the existing SLEUTH-3r application scope, we incorporated green space cores, as defined by MSPA, along with core importance and corridors into the exclusion layer of SLEUTH-3r that enabled us to generate urban growth scenarios. Adjusting the model's coefficients offered an additional approach to scenario generation, while assigning weights to urban zones produced scenarios for green space equity. We also considered limiting urban growth to 10% of the projected requirement as an alternative scenario, encouraging the development of high-rise buildings.

The social equity scenario resulted in the top 10 selected urbanization polygons being in the northern part of the study area. In this way, the already short coverage of green spaces in the eastern part will be saved from further deterioration, to the benefit of the communities living in this part. When using 23,850 ha of urbanization and including social equity, the urbanization polygons are distributed in a north and west of the study area. When including Exclusion 2 informed by the green spaces' cores, connectivity and corridor importance, the selected urbanization areas for the year 2050 became smaller and distributed in the landscape. To better protect green spaces' cores, their connectivity, and corridors while

**Fig 11. The generalized selected urbanization polygons for 2385 ha using usual and social equity scenarios.**

allowing for 23,850 ha of urban growth by the year 2050, the scenario "MSPA-Informed Compact-Social Equity" appears to be the best option using TOPSIS. However, for 2,385 hectares of urban growth, the 'Compact-Social Equity' scenario was identified as the best option by TOPSIS, influenced by the weights assigned to urbanization areas.

PARA_MN and ENN_MN values for the cores and urbanization polygons reflected the shape and proximity and were useful metrics for further investigation of the results. Since the main goal of this study was to develop and select the preferred scenarios for urban expansion while protecting green spaces using SLEUTH-3r, opting for slightly less weights for PARA_MN and ENN_MN of urban areas selected scenarios that had better values for green space cores and provided a sufficient urbanization by 2050. Many factors contribute to urban growth across the globe [72–80] and the trend of accelerated urban expansion is expected to continue in the foreseeable future [81], bringing about challenges such as the deterioration of green spaces. We generated scenarios of urban growth in Ottawa that incorporated green spaces. By comparing scenarios with and without green spaces, we highlighted the differences and provided a foundation for further exploration of potential combinations of input factors. This approach helps identify practical solutions for achieving green space-friendly urban growth. It also opens the door and encourages researchers to include explicit layers of environmental, economic and social aspects into their modeling tools such as SLEUTH-3r and enrich the results by providing scenarios to be selected by the urban planners and stakeholders.

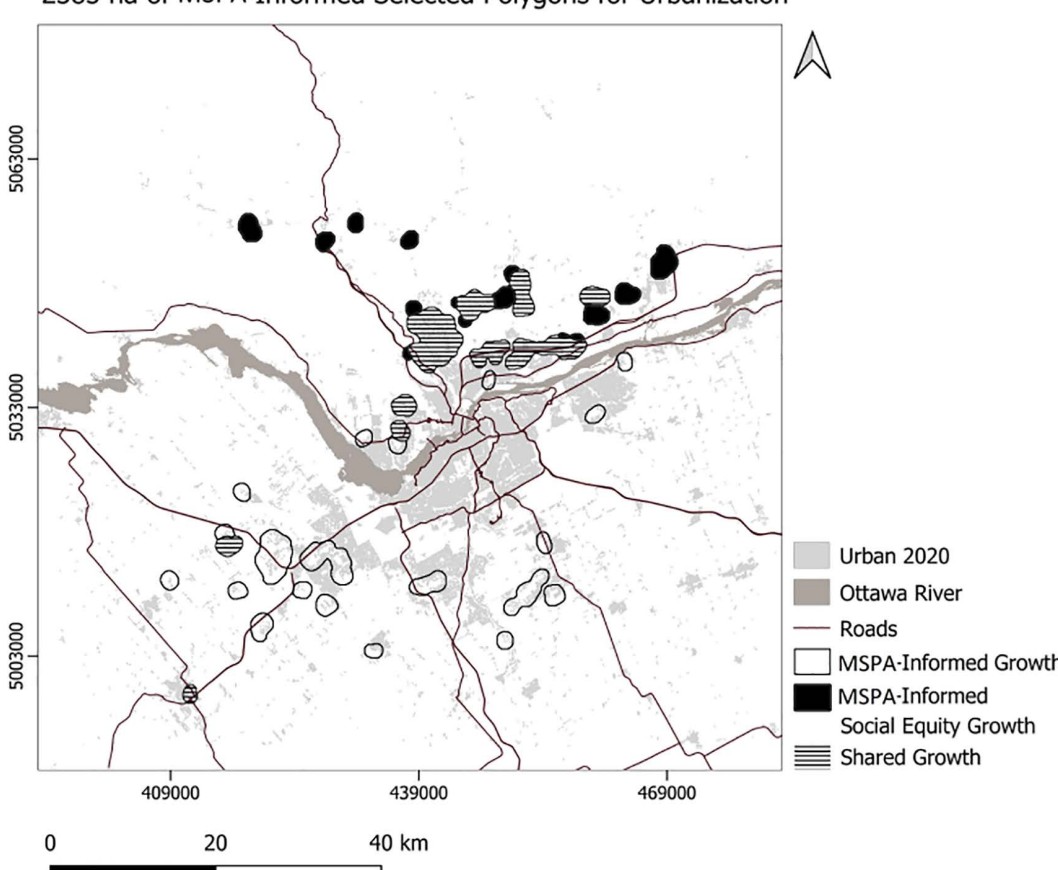

**Fig 12. MSPA-Informed, and social equity growth.**

## 5. Conclusion

Rapid urban expansion in recent decades and its continuation in the foreseeable future has introduced challenges such as the deterioration of green spaces, which can be mitigated through effective urban expansion planning. We used SLEUTH-3r to generate scenarios of urban expansion that also safeguard green spaces in Ottawa, Ontario, Canada. The scenarios were based on adjusting model coefficients to prioritize certain growth forms, employing two exclusion layers and assigning social equity weights to the results of SLEUTH-3r. These helped us create 8 scenarios of urban growth and assess their effects on green spaces. We then evaluated these scenarios and selected the optimal one using TOP-SIS. Our findings indicate that the inclusion of green space information significantly affects the results of the SLEUTH-3r model, providing a valuable basis for further on-the-ground evaluation and final decisions on suitable urban growth areas.

SLEUTH-3r requires extensive calibration effort, which is time-consuming and prone to human error. However, it also offers a platform for further exploration of various potential growth trajectories for the modeled urban area. In our study area, considering the size, urban ratio, and specifications of SLEUTH-3r, only the 90 by 90 m pixel size yielded acceptable results. The raster resolution of 90 by 90 m might not capture finer details in urban growth or green space changes. As this raster resolution provided the best model performance, further field-level studies are needed to incorporate finer details into the selected urban growth plans. Although incorporating slope, land use classes, areas excluded from urbanization, and roads in SLEUTH-3r implicitly suggests some urban suitability assessment, future studies could benefit from

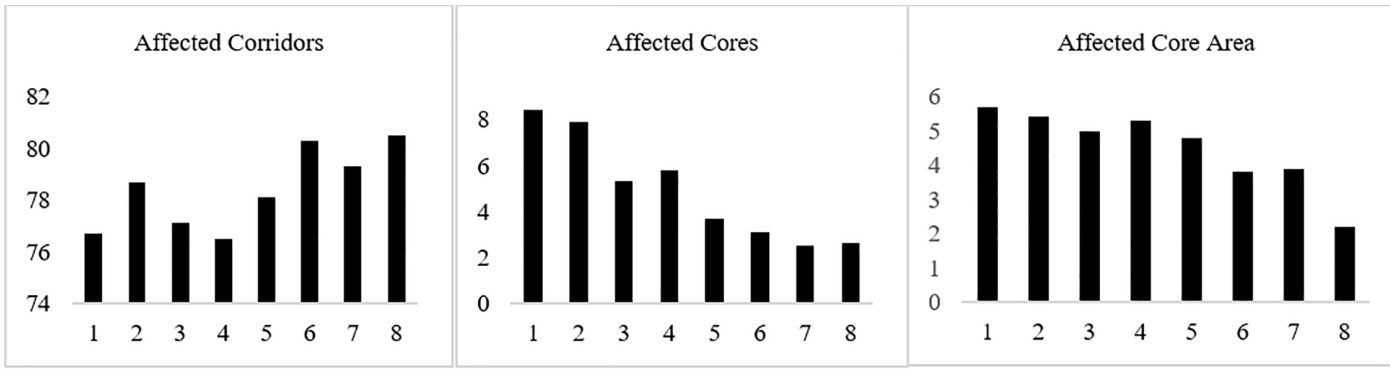

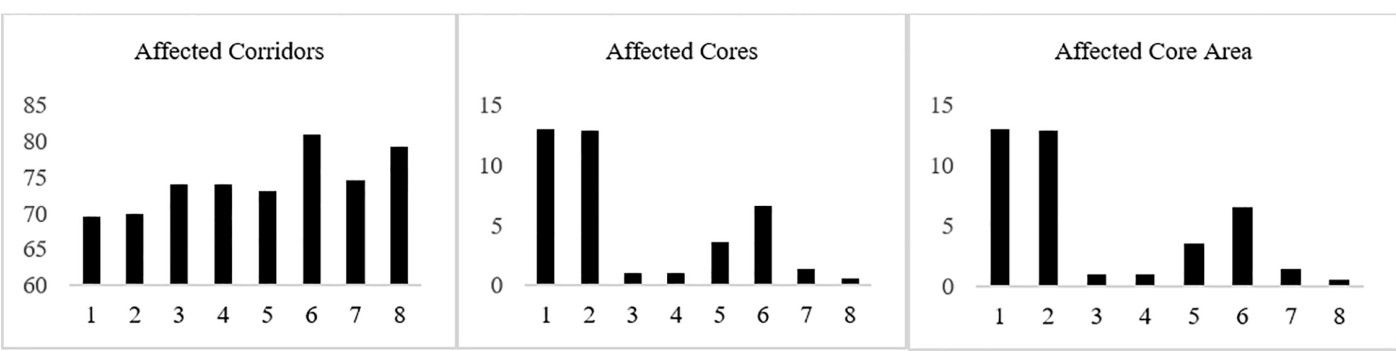

**Fig 13. Percentage of the affected high quality (>150) corridors and cores and the affected core area for 23850 ha (top row) and the best 2385 ha of the selected groups (bottom row).** X-Axis Legend: 1.Usual Growth, 2. Compact Growth, 3. Social Equity Growth, 4. Compact-Social Equity Growth, 5. MSPA-Informed Growth, 6. MSPA-Informed Compact Growth, 7. MSPA-Informed Social Equity Growth, 8. MSPA-Informed Compact-Social Equity Growth.

**Table 4. Features, values and weights used in the TOPSIS ranking of the scenarios.**

| Features | | Impacts on Corridors, Cores and Core Area | | | | | | Cores in 1 Km Buffer | | | | Selected Urbanization Areas | | | |
|---|---|---|---|---|---|---|---|---|---|---|---|---|---|---|---|
| | | Percent Affected Corridors>150 | | Percent Affected Cores>150 | | Percent Affected Core Area | | PARA_MN | | ENN_MN | | PARA_MN | | ENN_MN | |
| | Weights | 0.13 | | 0.2 | | 0.25 | | 0.12 | | 0.12 | | 0.06 | | 0.12 | |
| | Desirability | Min | | Min | | Min | | Min | | Min | | Min | | Min | |
| | Urbanization Area | 23850 ha | 2385 ha | 23850 ha | 2385 ha | 23850 ha | 2385 ha | 23850 ha | 2385 ha | 23850 ha | 2385 ha | 23850 ha | 2385 ha | 23850 ha | 2385 ha |
| Scenarios | Usual | 69.5 | 69.5 | 13 | 13 | 0.7 | 0.7 | 253.75 | 240.93 | 422.06 | 408.16 | 110.58 | 275.40 | 1614.78 | 1297.87 |
| | Compact | 69.9 | 69.9 | 12.9 | 12.9 | 0.66 | 0.66 | 250.09 | 240.49 | 424.64 | 407.76 | 109.42 | 151.94 | 1541.66 | 2833.66 |
| | Social Equity | 74 | 74 | 1 | 1 | 0.54 | 0.54 | 250.33 | 239.69 | 424.18 | 403.82 | 114.55 | 285.90 | 980.18 | 1468.18 |
| | Compact-Social Equity | 74 | 74 | 1 | 1 | 0.47 | 0.47 | 250.25 | 239.69 | 424.27 | 403.82 | 114.65 | 114.65 | 1045.36 | 1045.36 |
| | MSPA-Informed | 73 | 73 | 3.6 | 3.6 | 0.56 | 0.56 | 246.93 | 237.95 | 421.13 | 408.87 | 131.06 | 167.18 | 1089.63 | 2942.93 |
| | MSPA-Informed Compact | 81 | 81 | 6.6 | 6.6 | 0.4 | 0.4 | 245.92 | 238.35 | 417.68 | 408.69 | 148.09 | 155.59 | 1000.57 | 2500.26 |
| | MSPA-Informed Social Equity | 74.6 | 74.6 | 1.4 | 1.4 | 0.51 | 0.51 | 246.05 | 238.61 | 418.69 | 406.53 | 142.29 | 267.57 | 1110.93 | 1292.90 |
| | MSPA-Informed Compact-Social Equity | 79.3 | 79.3 | 0.6 | 0.6 | 0.44 | 0.44 | 243.27 | 238.65 | 414.34 | 406.93 | 164.89 | 287.30 | 987.73 | 2314.34 |

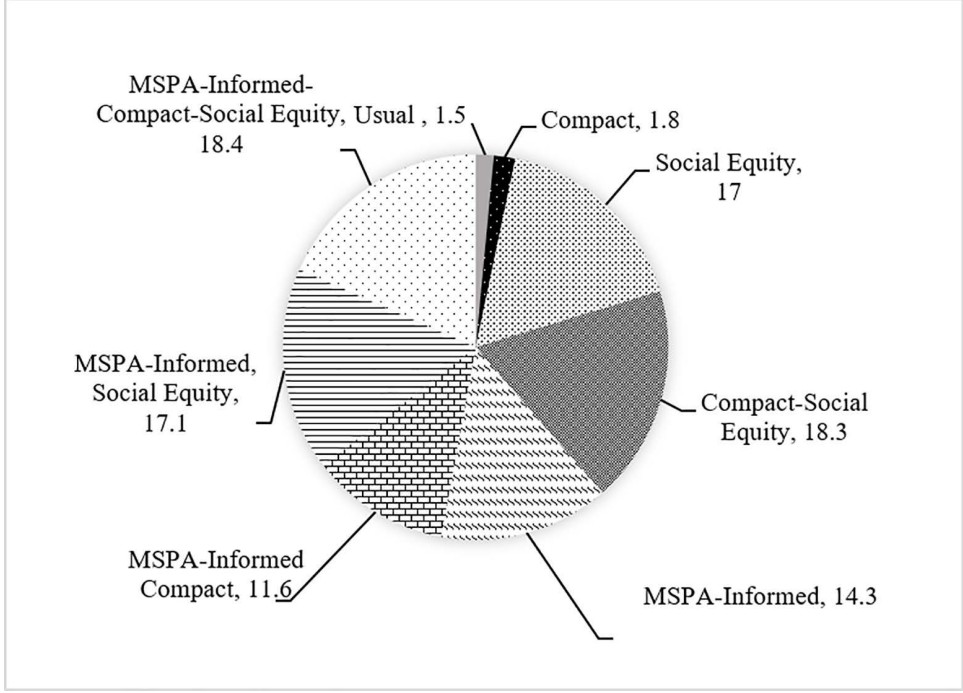

**Fig 14. TOPSIS results for all selected groups (23,850 ha).**

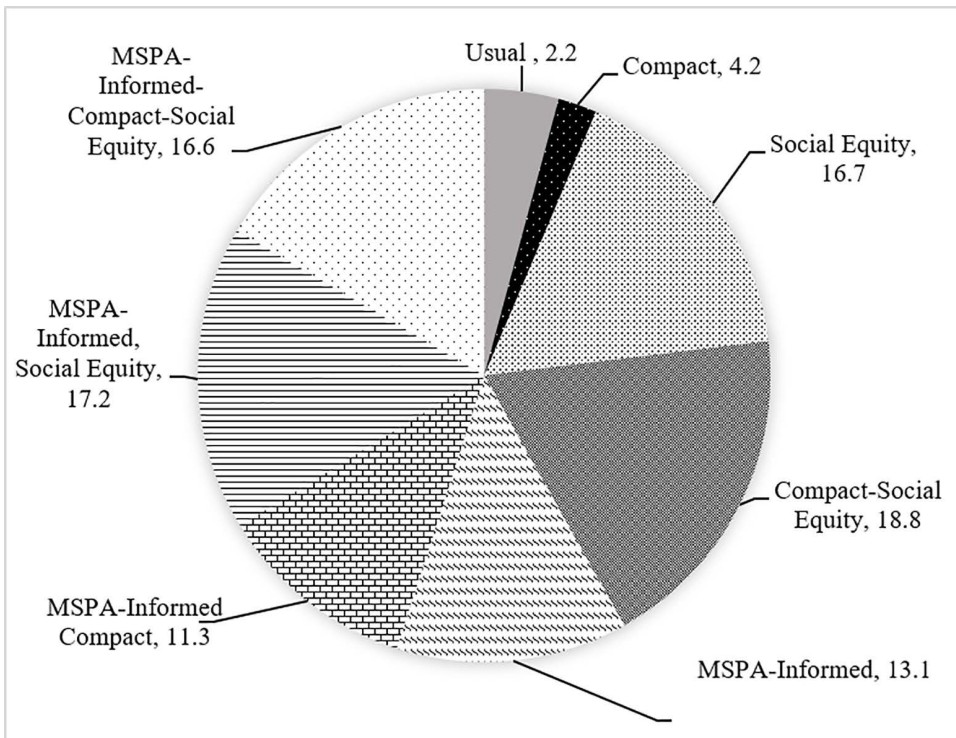

**Fig 15. TOPSIS results for the best 10 percent (2,385 ha).** Higher values indicate a higher rank for each scenario.

an explicit urban suitability evaluation. This should include factors such as distance to the town center and other important aspects of urban growth. Additionally, incorporating information on land prices, income and social group and other practical aspects of urban development could enhance our modeling.

In addition to the key finding that scenario generation using SLEUTH-3r can assist managers in balancing urban growth and green space protection, the study also presents a platform for exploring further scenarios. This approach allows for the incorporation of demographic, economic, environmental changes, and the dynamic nature of socio-political conditions into the model, facilitating a participatory approach to urban growth planning. As such, the best plans, based on the consensus of interest groups while safeguarding green spaces, can be selected for implementation.

## Supporting information

**S1 Fig. Slope layer.**
(TIF)

**S2 Fig. Hillshade layer.**
(TIF)

**S3 Fig. Land use layer for the year 1990.**
(TIF)

**S4 Fig. Land use layer for the year 2020.**
(TIF)

**S5 Fig. Urban extent layer for the year 1990.**
(TIF)

**S6 Fig. Urban extent layer for the year 2000.**
(TIF)

**S7 Fig. Urban extent layer for the year 2010.**
(TIF)

**S8 Fig. Urban extent layer for the year 2020.**
(TIF)

**S9 Fig. Road layer for the year 1990.**
(TIF)

**S10 Fig. Road layer for the year 2000.**
(TIF)

**S11 Fig. Road layer for the year 2010.**
(TIF)

**S12 Fig. Road layer for the year 2020.**
(TIF)

## Acknowledgments

We extend our sincere appreciation to Dr. Wade Hong, IT officer at Carleton University, for his invaluable assistance in facilitating the parallel running of SLEUTH-3r. Additionally, we would like to thank Dr. David I. Donato, Research Computer Scientist at USGS, for his invaluable support in running SLEUTH on HPC. We are also grateful for the assistance offered

by Dr. Claire Jantz, and Dr. Alfonso Yáñez Morillo, affiliated scholars with the Center for Land Use and Sustainability at Shippensburg University, USA. Map data copyrighted OpenStreetMap contributors and available from https://www.open-streetmap.org. We also thank Gorgan University of Agricultural Sciences and Natural Resources for providing the first author with the opportunity to conduct sabbatical studies. Finally, we express our gratitude to the anonymous reviewers for their valuable feedback.

**AI Use:** AI was used to help improve readability and language in the first draft of this paper. All text was subsequently reviewed and edited by all authors.

## Author contributions

**Conceptualization:** Abdolrassoul Salmanmahiny, Scott W. Mitchell, Joseph R. Bennett.

**Data curation:** Abdolrassoul Salmanmahiny, Scott W. Mitchell.

**Formal analysis:** Abdolrassoul Salmanmahiny, Scott W. Mitchell, Joseph R. Bennett.

**Funding acquisition:** Scott W. Mitchell, Joseph R. Bennett.

**Methodology:** Abdolrassoul Salmanmahiny, Scott W. Mitchell.

**Software:** Abdolrassoul Salmanmahiny.

**Writing – original draft:** Abdolrassoul Salmanmahiny.

**Writing – review & editing:** Abdolrassoul Salmanmahiny, Scott W. Mitchell, Joseph R. Bennett.

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
