## [Decision Letter · Decision Letter 0]

17 Jan 2025

PONE-D-24-60645
MSPA-Informed SLEUTH urban growth modeling for green space protection in Ottawa
PLOS ONE

Dear Dr. Salmanmahiny,

Thank you for submitting your manuscript to PLOS ONE. After careful consideration, we feel that it has merit but does not fully meet PLOS ONE’s publication criteria as it currently stands. Therefore, we invite you to submit a revised version of the manuscript that addresses the points raised during the review process.

We look forward to receiving your revised manuscript.

Kind regards,

Jun Yang

Academic Editor

PLOS ONE

Journal Requirements:

2. Please note that PLOS ONE has specific guidelines on code sharing for submissions in which author-generated code underpins the findings in the manuscript. In these cases, we expect all author-generated code to be made available without restrictions upon publication of the work. 

Please review our guidelines at https://journals.plos.org/plosone/s/materials-and-software-sharing#loc-sharing-code and ensure that your code is shared in a way that follows best practice and facilitates reproducibility and reuse.

“Part of the funding for the research presented in this paper was provided for the sabbatical studies of the first author by Gorgan University of Agricultural Sciences and Natural Resources, Gorgan, Iran. Additional funding was kindly provided by Carleton International and by the Department of Geography and Environmental Studies at Carleton University.”

“Part of the funding for the research presented in this paper was provided for the sabbatical studies of the first author by Gorgan University of Agricultural Sciences and Natural Resources, Gorgan, Iran. Additional funding was kindly provided by Carleton International and by the Department of Geography and Environmental Studies at Carleton University.”

5. Please note that funding information should not appear in the Acknowledgments section or other areas of your manuscript. We will only publish funding information present in the Funding Statement section of the online submission form. Please remove any funding-related text from the manuscript. 

6. We note that you have indicated that there are restrictions to data sharing for this study. PLOS only allows data to be available upon request if there are legal or ethical restrictions on sharing data publicly. For more information on unacceptable data access restrictions, please see http://journals.plos.org/plosone/s/data-availability#loc-unacceptable-data-access-restrictions. 

7. We note that your Data Availability Statement is currently as follows: 

“All relevant data are within the manuscript and its Supporting Information files.”

8. PLOS requires an ORCID iD for the corresponding author in Editorial Manager on papers submitted after December 6th, 2016. Please ensure that you have an ORCID iD and that it is validated in Editorial Manager. To do this, go to ‘Update my Information’ (in the upper left-hand corner of the main menu), and click on the Fetch/Validate link next to the ORCID field. This will take you to the ORCID site and allow you to create a new iD or authenticate a pre-existing iD in Editorial Manager.

9. We note that Figures 1 to 6 in your submission contain map/satellite images which may be copyrighted. All PLOS content is published under the Creative Commons Attribution License (CC BY 4.0), which means that the manuscript, images, and Supporting Information files will be freely available online, and any third party is permitted to access, download, copy, distribute, and use these materials in any way, even commercially, with proper attribution. For these reasons, we cannot publish previously copyrighted maps or satellite images created using proprietary data, such as Google software (Google Maps, Street View, and Earth). For more information, see our copyright guidelines: http://journals.plos.org/plosone/s/licenses-and-copyright.

1) You may seek permission from the original copyright holder of Figures 1 to 6 to publish the content specifically under the CC BY 4.0 license.  

2) If you are unable to obtain permission from the original copyright holder to publish these figures under the CC BY 4.0 license or if the copyright holder’s requirements are incompatible with the CC BY 4.0 license, please either i) remove the figure or ii) supply a replacement figure that complies with the CC BY 4.0 license. Please check copyright information on all replacement figures and update the figure caption with source information. If applicable, please specify in the figure caption text when a figure is similar but not identical to the original image and is therefore for illustrative purposes only.

**Additional Editor Comments:**

Reviewer 1

Authors must take care of the following suggestion for better understanding the article:

1. Complexity of Calibration: The SLEUTH-3r model requires extensive calibration with multiple parameters (Diffusion, Breed, Spread, etc.), which can make the process time-consuming and potentially prone to human error.

2. Assumptions on Growth Scenarios: The study relies on projected urbanization areas and assumes population growth patterns, which might deviate from reality due to unforeseen demographic, economic, or environmental changes.

3. Data Resolution: The raster resolution of 90x90 meters might not capture finer details in urban growth or green space changes, possibly overlooking critical micro-level variations.

4. Limited Suitability Assessment: While the model implicitly considers factors like slope and land use, it lacks a comprehensive urban suitability analysis that includes proximity to urban centers, infrastructure quality, or socio-economic factors.

Must follow the following articles for better assessment:

https://doi.org/10.1007/s10708-024-11240-1

https://doi.org/10.1016/j.ssaho.2024.101123

https://doi.org/10.1186/s13717-024-0533-5

https://doi.org/10.1007/s44243-023-00021-y

https://doi.org/10.1007/s41685-023-00313-7

https://doi.org/10.1016/j.regsus.2023.05.001

https://doi.org/10.1016/j.cstp.2023.100990

https://doi.org/10.1007/s10708-021-10571-7

5. Exclusion of Practical Constraints: Key factors such as land prices, zoning laws, and real-world policy constraints are not incorporated, which could affect the feasibility of the proposed scenarios.

6. Social Equity Weights: The assignment of social equity weights based on limited parameters (e.g., green spaces and population ratios) may not fully capture the complexity of urban socio-environmental dynamics.

7. Validation Limitations: While accuracy metrics like ROC and PR are used, the study could benefit from cross-validation with independent datasets or real-world case studies to enhance reliability.

8. Focus on Static Parameters: The analysis assumes static environmental and socio-political conditions, which may not reflect the dynamic nature of urban planning and green space management.

9. Subjectivity in TOPSIS Weight Assignment: The TOPSIS method, while systematic, is influenced by subjective weight assignments, which might bias the ranking of urban growth scenarios.

10. Limited Participatory Approach: The study suggests involving officials and stakeholders but does not explicitly incorporate participatory methodologies during scenario development or validation.

Reviewer 2

The authors explored the MSPA-Informed SLEUTH urban growth modeling for green space protection in Ottawa. The topic is interesting. However, the quality of writing is too low. The authors should illustrated your points clearly, and let readers understanding. The problems are as follow.

1.The keywords choosed simply. The core connectivity and core importance could be integrated. In general, keywords are phrase, rather than a word.

2.The introduction is really mess. The authors should introduced the insufficient of existing research and which gaps are you solved. In addition, the sentences should be shorted. For example, the content of line 97-101 could be illustrated that SLEUTH-3r has been applied in Baltimore Netherlands and China. Then compared their difference further. Listing the existing research is not allowed. For instance, line 80-93. The authors research......,the authors explored.....,it is lack of sumarizing.

3.The authors should introduced the reason for choosing this study area,rather than introduced basic information of this area simply.

4.The section 2.2 might could be integrated into section 2.3. The formula is also methods.

5.The authors should re-organized the text carefully. It should use less words and sentences to present your ideas clearly. It shouldn't listed the relevant sentence simply, the paper should emphasized the logic and readability.

6.The content of conclusion is like discussion. The discussion should compared difference of your new finding and existing research. Which aspects impoved the MSPA in your research? The conclusion should introduced your research and findings.

Reviewers' comments:

Reviewer's Responses to Questions

**Comments to the Author**

1. Is the manuscript technically sound, and do the data support the conclusions?

Reviewer #1: Yes

Reviewer #2: Yes

2. Has the statistical analysis been performed appropriately and rigorously? 

Reviewer #1: Yes

Reviewer #2: Yes

3. Have the authors made all data underlying the findings in their manuscript fully available?

Reviewer #1: Yes

Reviewer #2: Yes

4. Is the manuscript presented in an intelligible fashion and written in standard English?

Reviewer #1: Yes

Reviewer #2: Yes

5. Review Comments to the Author

Reviewer #1: Authors must take care of the following suggestion for better understanding the article:

1. Complexity of Calibration: The SLEUTH-3r model requires extensive calibration with multiple parameters (Diffusion, Breed, Spread, etc.), which can make the process time-consuming and potentially prone to human error.

2. Assumptions on Growth Scenarios: The study relies on projected urbanization areas and assumes population growth patterns, which might deviate from reality due to unforeseen demographic, economic, or environmental changes.

3. Data Resolution: The raster resolution of 90x90 meters might not capture finer details in urban growth or green space changes, possibly overlooking critical micro-level variations.

4. Limited Suitability Assessment: While the model implicitly considers factors like slope and land use, it lacks a comprehensive urban suitability analysis that includes proximity to urban centers, infrastructure quality, or socio-economic factors.

Must follow the following articles for better assessment:

https://doi.org/10.1007/s10708-024-11240-1

https://doi.org/10.1016/j.ssaho.2024.101123

https://doi.org/10.1186/s13717-024-0533-5

https://doi.org/10.1007/s44243-023-00021-y

https://doi.org/10.1007/s41685-023-00313-7

https://doi.org/10.1016/j.regsus.2023.05.001

https://doi.org/10.1016/j.cstp.2023.100990

https://doi.org/10.1007/s10708-021-10571-7

5. Exclusion of Practical Constraints: Key factors such as land prices, zoning laws, and real-world policy constraints are not incorporated, which could affect the feasibility of the proposed scenarios.

6. Social Equity Weights: The assignment of social equity weights based on limited parameters (e.g., green spaces and population ratios) may not fully capture the complexity of urban socio-environmental dynamics.

7. Validation Limitations: While accuracy metrics like ROC and PR are used, the study could benefit from cross-validation with independent datasets or real-world case studies to enhance reliability.

8. Focus on Static Parameters: The analysis assumes static environmental and socio-political conditions, which may not reflect the dynamic nature of urban planning and green space management.

9. Subjectivity in TOPSIS Weight Assignment: The TOPSIS method, while systematic, is influenced by subjective weight assignments, which might bias the ranking of urban growth scenarios.

10. Limited Participatory Approach: The study suggests involving officials and stakeholders but does not explicitly incorporate participatory methodologies during scenario development or validation.

Reviewer #2: The authors explored the MSPA-Informed SLEUTH urban growth modeling for green space protection in Ottawa. The topic is interesting. However, the quality of writing is too low. The authors should illustrated your points clearly, and let readers understanding. The problems are as follow.

1.The keywords choosed simply. The core connectivity and core importance could be integrated. In general, keywords are phrase, rather than a word.

2.The introduction is really mess. The authors should introduced the insufficient of existing research and which gaps are you solved. In addition, the sentences should be shorted. For example, the content of line 97-101 could be illustrated that SLEUTH-3r has been applied in Baltimore Netherlands and China. Then compared their difference further. Listing the existing research is not allowed. For instance, line 80-93. The authors research......,the authors explored.....,it is lack of sumarizing.

3.The authors should introduced the reason for choosing this study area,rather than introduced basic information of this area simply.

4.The section 2.2 might could be integrated into section 2.3. The formula is also methods.

5.The authors should re-organized the text carefully. It should use less words and sentences to present your ideas clearly. It shouldn't listed the relevant sentence simply, the paper should emphasized the logic and readability.

6.The content of conclusion is like discussion. The discussion should compared difference of your new finding and existing research. Which aspects impoved the MSPA in your research? The conclusion should introduced your research and findings.

6. PLOS authors have the option to publish the peer review history of their article (what does this mean?). If published, this will include your full peer review and any attached files.

Reviewer #1: No

Reviewer #2: No

---

## [Author Response · Author response to Decision Letter 1]

12 Feb 2025

We would like to thank the respected editor and reviewers for their thoughtful comments and suggestions, which have helped improve the quality of our manuscript. We have carefully considered all the editor’s and reviewers' comments, and below are our responses to them:

• We have provided both a highlighted and non-highlighted version of the revised manuscript.

• We referred to the PLOS ONE style templates to correct our manuscript.

• We uploaded our input data to ZENODO and provided the link at the end of the manuscript.

• We removed the funding information from the manuscript and included it in the cover letter, listing all relevant sources.

• We carefully reviewed the images for copyright concerns and removed the only one we suspected of being copyrighted (Fig 1, left). All others are free to use.

• We have addressed all comments to the best of our ability.

Reviewer #1:

Comments 1, 2, 3, 4. SLEUTH-3r being time-consuming and prone to human error, assuming static population growth patterns, deviating from reality due to unforeseen demographic, economic, or environmental changes, the raster resolution of 90x90 meters not capturing finer details in urban growth or green space changes should be mentioned. The suggested references should be considered.

Response: We have carefully mentioned the specifications and problems of modeling with SLEUTH-3r as suggested by the respected reviewer. We have also taken a look at the references suggested by the respected reviewer and in one case included a reference in our manuscript. These can be found in line 399 (reference 72) and lines 417-427.

Comment 5. Key and factors such as land prices, zoning laws, and real-world policy constraints not explicitly incorporated into the model.

Response: While a comprehensive inclusion of these factors in SLEUTH-3r are beyond the focus of the present manuscript, we have been aware of these research gaps waiting for future research. As such, we have mentioned these factors in lines 417-427.

Comment 6. The assignment of social equity weights based on limited parameters (e.g., green spaces and population ratios) may not fully capture the complexity of urban socio-environmental dynamics.

Response: Thank you for your valuable feedback. We agree with the reviewer’s comment and have already addressed this in lines 428-433, where we explain that our research offers a platform for further participatory trials. This approach allows users the flexibility to adjust the weights and incorporate additional factors into the process, enabling them to observe the results.

Comment 7. While accuracy metrics like ROC and PR are used, the study could benefit from cross-validation with independent datasets or real-world case studies to enhance reliability.

Response: We have taken great care to ensure the accuracy and validity of our results. In addition to OSM, we have included Precision-Recall alongside the ROC curve. We have also ensured validity using the Figure of Merit, as detailed in lines 231-245.

Comment 8. Assuming static environmental and socio-political conditions, may not reflect the dynamic nature of urban planning and green space management.

Response: We acknowledge this limitation in our research, as mentioned in lines 428-433, where we note that future studies can apply our method to incorporate new conditions and factors based on specific circumstances.

Comment 9. The TOPSIS method, while systematic, is influenced by subjective weight assignments, which might bias the ranking of urban growth scenarios.

Response: We acknowledge that the results in TOPSIS are influenced by the initial weights. However, as stated in lines 350-352, we iterated the calculations 20,000 times with a 50% variation in weights to account for differing perspectives, and this resulted in no significant changes to the outcomes. Additionally, in lines 428-433, we noted that our method can serve as a foundation for further exploration with new or alternative inputs.

Comment 10. The study suggests involving officials and stakeholders but does not explicitly incorporate participatory methodologies during scenario development or validation.

Response: While we recognize the importance of stakeholder involvement, explicitly implementing a participatory approach was beyond the scope of our research, which focused on SLEUTH-3r modeling and the incorporation of green space areas. However, in lines 428-433, we highlighted the potential for applying this approach in future studies.

Reviewer #2:

Comment 1. Keywords should be made simpler.

Response: We have shortened a few of the keywords where possible.

Comment 2. Simplification, clarification and summarizing is needed for the introduction section.

Response: Significant revisions have been made to the introduction, including the removal and shortening of some sentences for clarity. In lines 117-128, the justification and research goals are now presented in a more direct and concise manner.

Comment 3. The reason for choosing the study area should be mentioned.

Response: We have provided a sentence in lines 137-140 explaining the reason for choosing the study area.

Comment 4. Moving some sentences from the results to methods is needed.

Response: Thank you. We have moved the core importance and corridor assessment methods and formulae from results section to methods section, now in lines 189-200.

Comment 5. Reorganization of the text and more clarification and readability is required.

Response: Extensive revisions have been made throughout the text to enhance clarity and readability. We hope that the respected reviewer and readers will find these improvements beneficial.

Comment 6. The conclusion section should provide the main findings of the research.

Response: We have carefully addressed this comment in the conclusion section by removing references, presenting the findings more clearly, and acknowledging the study's limitations. We hope the respected reviewer and readers will find these improvements valuable.

---

## [Decision Letter · Decision Letter 1]

16 Mar 2025

PONE-D-24-60645R1
MSPA-Informed SLEUTH urban growth modeling for green space protection in Ottawa
PLOS ONE

Dear Dr. Salmanmahiny,

Thank you for submitting your manuscript to PLOS ONE. After careful consideration, we feel that it has merit but does not fully meet PLOS ONE’s publication criteria as it currently stands. Therefore, we invite you to submit a revised version of the manuscript that addresses the points raised during the review process.

We look forward to receiving your revised manuscript.

Kind regards,

Jun Yang

Academic Editor

PLOS ONE

Additional Editor Comments:

Major Revision

Reviewers' comments:

Reviewer's Responses to Questions

**Comments to the Author**

1. If the authors have adequately addressed your comments raised in a previous round of review and you feel that this manuscript is now acceptable for publication, you may indicate that here to bypass the “Comments to the Author” section, enter your conflict of interest statement in the “Confidential to Editor” section, and submit your "Accept" recommendation.

Reviewer #2: All comments have been addressed

Reviewer #3: (No Response)

2. Is the manuscript technically sound, and do the data support the conclusions?

Reviewer #2: Yes

Reviewer #3: Yes

3. Has the statistical analysis been performed appropriately and rigorously? 

Reviewer #2: Yes

Reviewer #3: Yes

4. Have the authors made all data underlying the findings in their manuscript fully available?

Reviewer #2: Yes

Reviewer #3: Yes

5. Is the manuscript presented in an intelligible fashion and written in standard English?

Reviewer #2: Yes

Reviewer #3: Yes

6. Review Comments to the Author

Reviewer #2: Although the authors have revised manuscript carefully, there still have problems need to be revised. The comments are as follow.

1.The innovation and contribution of the paper need to be further highlighted.

Although the paper mentioned tools such as SLEUTH-3r and MSPA, these methods are not novel. The paper's innovations need to be further emphasized, such as: Is it the first time that MSPA core importance and connectivity are integrated into the SLEUTH-3r model? What is the progress or uniqueness of this study compared with existing studies (e.g., Is it also suitable apllied in other cities)?

2.The research hypothesis and objectives are not clear enough.

Although the background of the study is introduced in the introduction section, the hypothesis and specific objectives of the study are not clearly stated. It is suggested to supplement the following contents:

The main assumptions of the study (e.g., can green space protection be achieved through scenario optimization?).

Clear research objectives (such as proposing green space protection strategies applicable to rapidly urbanized areas).

3. The data sources and processing are not clear.

Although the data source section is comprehensive, some of the details of data processing are not clear enough.

For example, how to ensure the consistency between different data sources according to GLC_CFS30, Dynamic World and GAIA classification standards and reclassification details?

The processing of road data is mentioned to be manually deleted through Google Earth, but this subjective method may bring some errors, so it is suggested to discuss its limitations.

4.The construction of the exclusion layer requires more detailed description.

Exclusion layers Exclusion 1 and Exclusion 2 are the key to the study, but the paper describes their construction process vaguely. For example, how to quantify "higher core importance of green space" and "high current area"?

Exclusion 2 The parameter setting of MSPA classification and Conefor calculation in Exclusion 2 should be more detailed.

The differences between the two exclusion layers and their specific impact on the results need to be discussed in depth.

5.There are deficiencies in the model calibration and verification section.

Although the calibration of SLEUTH-3r is mentioned to use OSM indicators, the calibration steps and parameter Settings are not explained in detail. It is suggested to supplement the weight and selection basis of each index in the calibration process. The validation section uses only ROC and PR indicators, without explaining why these indicators were chosen or discussing the potential uncertainty of model predictions. It is suggested to discuss the applicability of ROC and PR results, and whether there are other indicators that can be supplemented (such as Kappa coefficient).

6.The limitations of scenario analysis is insufficient in discussion

Although the paper puts forward 8 scenarios, it does not discuss the limitations and applicability of each scenario.

For example, will "compact growth" lead to social and economic imbalances in some regions? How scientific is the distribution basis and adjustment method of social equity weight? It is suggested to supplement the discussion.

7. The figures and tables questions:

Some figures lack clear explanations (such as the meaning of colors and symbols in Figure 4-6), which may lead to difficulties in understanding. The numerical comparison in table 3 is clear, but lacks a prominent description of key results (such as core data for the optimal scenario). It is recommended to add more explanatory text in the chart description and clearly emphasize the conclusions of key charts in the text.

8.The keywords are too much. I suggest remove the Ontario,Scenario.

9.In line 92, the authors mentioned the appropriate resolution of input images for the model to function effectively remains a research gap. Is this gap solved in your research?

10.The sub-title 2.1 should be revised to study area. The 2.2 should be revised to Data source. The 2.3 could revised to Methods.

11.In line 134-135, this area has recently undergone rapid urban growth, threatening green spaces and justifying the need to focus on future city growth projections and effective management strategies. It should be illustrated detailed further. Such as supply data.

12.I have a question, the earliest OSM street is 2014, how to get 1900,2000,2010 road network? It should be illustrated clearly.

13.The logic is a little mess. Such as the content of SLEUTH-3r could be integrated in method section. The content of scenarios also could be integrated. I suggest add a technique map in method section to make it clear. In line 330, a sentence as a paragraph is not suitable.

14.The content should corresponding to specific figures or tables. Such as line 283-287, and line 305-310.

15.The language and expression problems.

Some sentences are too long and not concise enough, such as the introduction and methods sections.

It is recommended to further polish the language to ensure that the expression is concise and logical.

16.The references cited are not standard enough. For example, the section 2.2 data source, this section is not necessary cited references. The references should not repeat shown many times in this manuscript such as line 212 and 213, the number should be in sequence，like 1,2,3....50. The citation in line 156-157,165 also not necessary.

17.Some important references should be cited in line 59-60 as follow.

Spatiotemporal patterns of vegetation phenology along the urban-rural gradient in Coastal Dalian, China.Urban Forestry & Urban Greening,2020,54,126784.doi:10.1016/j.ufug.2020.126784

Spatiotemporal variation characteristics of green space ecosystem service value at urban fringes: A case study on Ganjingzi District in Dalian, China.Science of The Total Environment,2018,639:1453-1461.doi: https://doi.org/10.1016/j.scitotenv.2018.05.253

Spatial and temporal heterogeneity of urban land area and PM2.5 concentration in China. Urban Climate,2022,45:101268. doi: https://doi.org/10.1016/j.uclim.2022.101268.

Spatial influence of exposure to green spaces on the climate comfort of urban habitats in China. Urban Climate,2023,51:101657. doi: https://doi.org/10.1016/j.uclim.2023.101657.

Reviewer #3: This article comprehensively utilizes the SLEUTH-3r model, MSPA method, and TOPSIS method to study how to protect Ottawa's green spaces in the context of urban expansion, providing insights into future social equity and compact city scenarios, and achieving interesting results that serve as an important reference for local governments. My suggestions are as follows:

Further emphasize the article's innovation points. Clearly articulate the article's innovations in the introduction or discussion section and supplement existing related research more extensively. For example, the authors could reference the role of rooftop greening and pocket parks (10.1016/j.scs.2025.106261, https://doi.org/10.1016/j.foar.2023.12.007) as well as the benefits of protecting green corridors for urban ecology (10.3390/land11020165, 10.1016/j.apgeog.2024.103439).

Is collecting land use data every ten years for prediction too coarse? In the methods section, provide reasons for the selected time points, such as data availability and the validity of the time span for urban growth prediction. Would more detailed temporal classifications improve model accuracy? How would using higher-resolution data (e.g., 30 meters or 10 meters) affect model results? Could future research be suggested to verify this?

What is the definition of a compact city scenario? How can it be more accurately quantified? Suggest adding a clear definition of a compact city in Section 2.3.7, for example: "A compact city emphasizes high-density development, mixed land use, and efficient transportation systems." Provide specific quantitative indicators, such as building density (building area per hectare), population density (population per square kilometer), spatial proportion, high-rise building proportion, or average building height. Explain the rationale for simulating compact cities by adjusting diffusion, breed, road gravity, and slope resistance coefficients in the SLEUTH-3r model.

How are the weights in the TOPSIS method assigned? In Section 2.3.9, describe the weight allocation method in detail and explain the rationale for assigning weights to each indicator, such as why the affected core area is given the highest weight. Compare this with other methods like the Analytic Hierarchy Process (AHP) or entropy weight method. The authors mention sensitivity analysis of weights (e.g., the 20,000 iterations mentioned in the article); please further explain its impact on the results.

What is the rationale for selecting green space cores (including cores, perforations, and edges)? Is there sufficient justification? Suggest supplementing the explanation of the MSPA classification method and its application in selecting green space cores in Section 2.3.1. Cite relevant literature to support the rationale for selecting cores, perforations, and edges, such as their significance to ecosystem connectivity. Based on the evaluation results from Conefor software, explain the significance of core selection for overall green space protection.

In the conclusion section, expand the discussion on future research directions, such as how to optimize the model by incorporating more socio-economic factors and policy changes. Further clarify the practical application value of the article, such as providing scientific guidance for urban planners and policymakers, for example, what protection strategies should be adopted in different regions and whether these strategies have room for dynamic adjustments.

7. PLOS authors have the option to publish the peer review history of their article (what does this mean?). If published, this will include your full peer review and any attached files.

Reviewer #2: No

Reviewer #3: No

---

## [Author Response · Author response to Decision Letter 2]

14 Apr 2025

We would like to thank the respected editor and reviewers for their thoughtful comments and suggestions, which have helped improve the quality of our manuscript. We have carefully considered all the editor’s and reviewers' comments, and below are our responses to them:

• We have provided both a highlighted and non-highlighted version of the revised manuscript.

• We have addressed all comments to the best of our ability.

Reviewer #2:

Comment 1. The innovation and contribution of the paper need to be further highlighted. The paper's innovations need to be further emphasized, such as: Is it the first time that MSPA core importance and connectivity are integrated into the SLEUTH-3r model?

Response:

As far as the length of the paper allowed, the innovation and uniqueness of the study are highlighted in lines 46-47, 107-109 and 122-126.

Comment 2. The research hypothesis and objectives are not clear enough. It is suggested to supplement the following contents: The main assumptions of the study (e.g., can green space protection be achieved through scenario optimization?). Clear research objectives (such as proposing green space protection strategies applicable to rapidly urbanized areas).

Response:

The required sentences as to the objectives and assumptions of the study have been added to the paper in lines 113-119.

Comment 3. The data sources and processing are not clear. For example, how to ensure the consistency between different data sources according to GLC_CFS30, Dynamic World and GAIA classification standards and reclassification details?

Response:

Regarding consistency between data sources, lines 144–147 of the paper explain that only the urban areas from Dynamic World and GAIA were used to supplement and compare the urban areas in the GLC-CFS30 dataset. Therefore, it was not necessary to align the different classification schemes used by these sources.

Comment 4. The construction of the exclusion layer requires more detailed description. Exclusion layers Exclusion 1 and Exclusion 2 are the key to the study, but the paper describes their construction process vaguely. The differences between the two exclusion layers and their specific impact on the results need to be discussed in depth.

Response:

In lines 173-177, we have provided more detailed explanation of the differences between the two exclusion layers.

Comment 5. Although the calibration of SLEUTH-3r is mentioned to use OSM indicators, the calibration steps and parameter Settings are not explained in detail. It is suggested to supplement the weight and selection basis of each index in the calibration process. It is suggested to discuss the applicability of ROC and PR results, and whether there are other indicators that can be supplemented (such as Kappa coefficient).

Response:

As far as the length of the paper allowed, more information on calibration steps has been added in lines 235-242. Although there are other methods to assess the accuracy of the results, we have mentioned in lines 247-253 that in our case, using ROC and PR are more appropriate than Kappa coefficient.

Comment 6. Although the paper puts forward 8 scenarios, it does not discuss the limitations and applicability of each scenario. For example, will "compact growth" lead to social and economic imbalances in some regions? How scientific is the distribution basis and adjustment method of social equity weight? It is suggested to supplement the discussion.

Response:

In lines 264-275 of the paper, we have added explanation of how we have generated compact growth and its implications. In lines 280-287, and 289-290 we have further explained the process and the need to future studies as to the imbalances that may occur as a result of putting the compact city and social equity scenarios into effect.

Comment 7. Some figures lack clear explanations (such as the meaning of colors and symbols in Figure 4-6), which may lead to difficulties in understanding. The numerical comparison in table 3 is clear, but lacks a prominent description of key results.

Response:

In lines 351-358, we have added explanation of the meaning of colors and symbols in Figures 4-7 (old numbers: Figures 4-6). Also, in lines 318-323, more description has been given for key results of Table 3.

Comment 8. The keywords are too much. I suggest remove the Ontario, Scenario.

Response:

Two keywords have been removed.

Comment 9. In line 92, the authors mentioned the appropriate resolution of input images for the model to function effectively remains a research gap. Is this gap solved in your research?

Response:

In lines 310-311 and 461-464 of the paper, we have explained the appropriate resolution of the input images for our study area. Further in-depth analysis of this aspect requires more technical explorations to be covered in future studies.

Comment 10. The sub-title 2.1 should be revised to study area. The 2.2 should be revised to Data source. The 2.3 could revised to Methods.

Response:

The required revisions have been implemented.

Comment 11. In line 134-135, this area has recently undergone rapid urban growth, threatened green spaces and justified the need to focus on future city growth projections and effective management strategies. It should be supplied with data.

Response:

In line 137, we supplied quantitative data as to the rapid urban growth that has occurred in our study area.

Comment 12. I have a question, the earliest OSM street is 2014, how to get 1990,2000,2010 road network? It should be illustrated clearly.

Response:

Thanks for your comment. In lines 179-186, we have added further explanation as to how the road layers for the earlier years have been generated.

Comment 13. I suggest to add a technique map in method section to make it clear. In line 330, a sentence as a paragraph is not suitable.

Response:

We have added a flowchart map of the study as Figure 4. We hope this figure provides clear description of the steps taken in our study.

Comment 14. The content should correspond to specific figures or tables. Such as line 283-287, and line 305-310.

Response:

Thanks for your comment. We have addressed these issues.

Comment 15. The language and expression problems. Some sentences are too long and not concise enough, such as the introduction and methods sections. It is recommended to further polish the language to ensure that the expression is concise and logical.

Response:

We have tried to address this issue by making some the long sentences shorter and polishing the language. Hope the paper is now more readable.

Comment 16. The references cited are not standard enough. For example, the section 2.2 data source, this section is not necessary cited references. The references should not repeat shown many times in this manuscript such as line 212 and 213, the number should be in sequence like 1,2,3....50. The citation in line 156-157,165 also not necessary.

Response:

We have removed the unnecessary references and their repetitions in the paper.

Comment 17. Some important references should be cited in line 59-60 as follow. Spatiotemporal patterns of vegetation phenology along the urban-rural gradient in Coastal Dalian, China. Urban Forestry &Urban Greening, 2020,54,126784. doi:10.1016/j.ufug.2020.126784, Spatiotemporal variation characteristics of green space ecosystem service value at urban fringes: A case study on Ganjingzi District in Dalian, China. Science of The Total Environment, 2018,639:1453-1461.doi: https://doi.org/10.1016/j.scitotenv.2018.05.253, Spatial and temporal heterogeneity of urban land area and PM2.5 concentration in China. UrbanClimate,2022,45:101268. doi: https://doi.org/10.1016/j.uclim.2022.101268. Spatial influence of exposure to green spaces on the climate comfort of urban habitats in China. UrbanClimate,2023,51:101657. doi: https://doi.org/10.1016/j.uclim.2023.10

Response:

Thanks for your comment. We have now included the above references numbered 6-9 in our paper.

Reviewer #3:

Comment 1. Further emphasize the article's innovation points in the introduction or discussion section and supplement existing related research more extensively. For example, the authors could reference the role of rooftop greening and pocket parks (10.1016/j.scs.2025.106261, https://doi.org/10.1016/j.foar.2023.12.007) as well as the benefits of protecting green corridors for urban ecology (10.3390/land11020165,10.1016/j.apgeog.2024.103439).

Response:

As far as the length of the paper allowed, the innovation and uniqueness of the study are highlighted in lines 46-47, 107-109 and 122-126. We have also included the mentioned references in our paper, now numbered 58-61.

Comment 2. Is collecting land use data every ten years for prediction too coarse? In the methods section, provide reasons for the selected time points, such as data availability and the validity of the time span for urban growth prediction. Would more detailed temporal classifications improve model accuracy?

Response:

As explained in lines 147–149, the 10-year interval for the input images was selected primarily for two reasons: the availability of data and the need for a sufficient period to capture detectable urban growth in the imagery.

Comment 3. How would using higher-resolution data (e.g., 30 meters or 10meters) affect model results? Could future research be suggested to verify this?

Response:

In lines 310-311 and 461-464 of the paper, we have explained the appropriate resolution of the input images for our study area. Further in-depth analysis of this aspect requires more technical explorations to be covered in future studies. We have mentioned this need in lines 461-465.

Comment 4. What is the definition of a compact city scenario? How can it be more accurately quantified? Suggest adding a clear definition of a compact city in Section 2.3.7, for example: "A compact city emphasizes high-density development, mixed land use, and efficient transportation systems."

Response:

In lines 264-275, we have explained “Compact Scenario”. Thanks for your comment.

Comment 5. Explain the rationale for simulating compact cities by adjusting diffusion, breed, road gravity, and slope resistance coefficients in the SLEUTH-3r model.

Response:

In lines 264-275, we have explained the rationale for simulating Compact Scenario by adjusting the SLEUTH-3r coefficients.

Comment 6. How are the weights in the TOPSIS method assigned? In Section 2.3.9, describe the weight allocation method in detail and explain the rationale for assigning weights to each indicator, such as why the affected core area is given the highest weight. Compare this with other methods like the Analytic Hierarchy Process (AHP) or entropy weight method.

Response:

In lines 377-389 we have added more explanation of the weights used in the TOPSIS and the rationale behind these decisions. Also, more information on the preferences given to the green space cores has been provided in these lines.

Comment 7. The authors mention sensitivity analysis of weights (e.g., the 20,000 iterations mentioned in the article); please further explain its impact on the results.

Response:

As far as the length of the paper allowed, more information on sensitivity analysis in the TOPSIS has been provided in lines 377-389. Thanks for your comment.

Comment 8. What is the rationale for selecting green space cores (including cores, perforations, and edges)? Is there sufficient justification? Suggest supplementing the explanation of the MSPA classification method and its application in selecting green space cores in Section 2.3.1. Cite relevant literature to support the rationale for selecting cores, perforations, and edges, such as their significance to ecosystem connectivity.

Response:

In lines 96-98 and 160-161, more explanation has been provided on MSPA and the rationale for selecting green space cores. In these lines, we have also provided due reference justifying our practice.

Comment 9. In the conclusion section, expand the discussion on future research directions, such as how to optimize the model by incorporating more socio-economic factors and policy changes. Further clarify the practical application value of the article, such as providing scientific guidance for urban planners and policymakers, for example, what protection strategies should be adopted in different regions and whether these strategies have room for dynamic adjustments.

Response:

In lines 464-470, we have provided discussion on future research directions and in lines 471-477, we have indicated the applications of the research. Thanks for your comment.

Once again, we appreciate the reviewers' detailed feedback and believe the revisions have strengthened the manuscript. We hope these changes meet the reviewers' expectations. We believe our manuscript will provide useful information on application of SLEUTH-3r and inclusion of green space information into the process, especially for large areas with dispersed urban growth. We are happy to make further changes if the esteemed reviewers deem them necessary.

Best regards

The authors

---

## [Decision Letter · Decision Letter 2]

5 May 2025

PONE-D-24-60645R2
MSPA-Informed SLEUTH urban growth modeling for green space protection in Ottawa
PLOS ONE

Dear Dr. Salmanmahiny,

Thank you for submitting your manuscript to PLOS ONE. After careful consideration, we feel that it has merit but does not fully meet PLOS ONE’s publication criteria as it currently stands. Therefore, we invite you to submit a revised version of the manuscript that addresses the points raised during the review process.

We look forward to receiving your revised manuscript.

Kind regards,

Jun Yang

Academic Editor

PLOS ONE

Journal Requirements:

Additional Editor Comments:

The manuscript have been improved. However, there still have some problems need to be revised.

1.Although the paper the first mentions combination of MSPA and SLEUTH-3r, it does not fully elaborated its theoretical contribution. For example, how to expand the existing SLEUTH-3r application scope, or how to fill the gap of MSPA in urban expansion modeling?

2.The SLEUTH-3r calibration step has been added, there is still a lack of detailed discussion on how to adjust the key parameters (such as Diffusion, Breed, Road Gravity, etc.), especially how to derive these parameters from actual geographical conditions.

3.The paper mentions that "compact cities" may lead to social and economic imbalances, but the specific suggestion are not mentioned. For example, how different income groups will be affected?

4.It is lack of green space protection policy of different regions (such as urban core area and suburbs), and there is a lack of more targeted practical guidance.

5.Some paragraphs in the paper are too long (such as the method section), which affects the fluency of reading. For example, Introduction and Methods sections can be further compressed to highlight the key points.

6.The symbol interpretation of some charts (such as Figure 4-6) is still brief, and the spatial differences of different scenarios are not intuitively reflected.

Reviewers' comments:

Reviewer's Responses to Questions

**Comments to the Author**

1. If the authors have adequately addressed your comments raised in a previous round of review and you feel that this manuscript is now acceptable for publication, you may indicate that here to bypass the “Comments to the Author” section, enter your conflict of interest statement in the “Confidential to Editor” section, and submit your "Accept" recommendation.

Reviewer #2: All comments have been addressed

Reviewer #3: All comments have been addressed

2. Is the manuscript technically sound, and do the data support the conclusions?

Reviewer #2: Yes

Reviewer #3: Yes

3. Has the statistical analysis been performed appropriately and rigorously? 

Reviewer #2: Yes

Reviewer #3: Yes

4. Have the authors made all data underlying the findings in their manuscript fully available?

Reviewer #2: Yes

Reviewer #3: Yes

5. Is the manuscript presented in an intelligible fashion and written in standard English?

Reviewer #2: Yes

Reviewer #3: Yes

6. Review Comments to the Author

Reviewer #2: The manuscript have been improved. However, there still have some problems need to be revised.

1.Although the paper the first mentions combination of MSPA and SLEUTH-3r, it does not fully elaborated its theoretical contribution. For example, how to expand the existing SLEUTH-3r application scope, or how to fill the gap of MSPA in urban expansion modeling?

2.The SLEUTH-3r calibration step has been added, there is still a lack of detailed discussion on how to adjust the key parameters (such as Diffusion, Breed, Road Gravity, etc.), especially how to derive these parameters from actual geographical conditions.

3.The paper mentions that "compact cities" may lead to social and economic imbalances, but the specific suggestion are not mentioned. For example, how different income groups will be affected?

4.It is lack of green space protection policy of different regions (such as urban core area and suburbs), and there is a lack of more targeted practical guidance.

5.Some paragraphs in the paper are too long (such as the method section), which affects the fluency of reading. For example, Introduction and Methods sections can be further compressed to highlight the key points.

6.The symbol interpretation of some charts (such as Figure 4-6) is still brief, and the spatial differences of different scenarios are not intuitively reflected.

Reviewer #3: This article has addressed all my questions. I believe it has met the standards for journal publication, and I have no further comments.

7. PLOS authors have the option to publish the peer review history of their article (what does this mean?). If published, this will include your full peer review and any attached files.

Reviewer #2: No

Reviewer #3: No

---

## [Author Response · Author response to Decision Letter 3]

21 Jun 2025

Answers to the reviewers’ comments, Third Round

We would like to thank the respected editor and reviewers for their thoughtful comments and suggestions, which have helped improve the quality of our manuscript. We have carefully considered all the editor’s and reviewers' comments, and below are our responses to them:

• We have provided both a highlighted and non-highlighted version of the revised manuscript.

• We have addressed all comments to the best of our ability.

Reviewer #2:

Comment 1. Although the paper mentions the first combination of MSPA and SLEUTH-3r, it does not fully elaborate how to expand the existing SLEUTH-3r application scope, or how to fill the gap of MSPA in urban expansion modeling?

Response:

Within the constraints of the paper’s length, references to expanding the scope of the SLEUTH-3r application were added in lines 87–88, 94–95, and 420–422.

Comment 2. The SLEUTH-3r calibration step has been added, there is still a lack of detailed discussion on how to adjust the key parameters (such as Diffusion, Breed, Road Gravity, etc.), especially how to derive these parameters from actual geographical conditions.

Response:

Within the allowed length of the paper, we included a description of how to adjust the coefficients of the SLEUTH-3r model in lines 218–225.

Comment 3. The paper mentions that "compact cities" may lead to social and economic imbalances, but the specific suggestions are not mentioned. For example, how different income groups will be affected?

Response:

The effects of model results on income groups warrant separate future studies and fall beyond the scope of this paper. However, we have acknowledged these limitations in the Methods, Discussion, and Conclusion sections (lines 274, 450–451, and 475–477) to guide interested researchers.

Comment 4. It is lack of green space protection policy of different regions (such as urban core area and suburbs), and there is a lack of more targeted practical guidance.

Response:

Regarding practical guidance, we believe the maps generated in our study offer a strong foundation for supporting green space protection in Ottawa’s urban planning. However, we acknowledge that detailed, on-the-ground decisions require finer-scale data and consideration of additional factors. Our results currently suggest that urban growth is more suitable along the northeast–southwest axis of the region. We have elaborated on this in lines 45–46, 358–369, and 411–412.

Comment 5 Some paragraphs in the paper are too long (such as the method section), which affects the fluency of reading. For example, Introduction and Methods sections can be further compressed to highlight the key points.

Response:

We have revised and shortened the paragraphs in the Introduction and Methods sections to improve clarity and readability.

Comment 6. The symbol interpretation of some charts (such as Figure 5-7) is still brief, and the spatial differences of different scenarios are not intuitively reflected.

Response:

We have clarified the interpretation of Figures 5–7 and explained the symbols used in lines 358–369 of the manuscript.

Reviewer #3:

No further comments have been suggested by the respected Reviewer #3.

Once again, we sincerely appreciate the reviewers’ detailed feedback, which we believe has significantly improved the manuscript. We hope the revisions meet the reviewers’ expectations. Our study offers valuable insights into the application of SLEUTH-3r and the integration of green space data, particularly for large regions experiencing dispersed urban growth. We respectfully hope the revised manuscript will be found suitable for publication in your esteemed journal.

Best regards

The authors

---

## [Decision Letter · Decision Letter 3]

4 Jul 2025

MSPA-Informed SLEUTH urban growth modeling for green space protection in Ottawa

PONE-D-24-60645R3

Dear Dr. Salmanmahiny,

We’re pleased to inform you that your manuscript has been judged scientifically suitable for publication and will be formally accepted for publication once it meets all outstanding technical requirements.

Kind regards,

Jun Yang

Academic Editor

PLOS ONE

Additional Editor Comments (optional):

Accept

Reviewers' comments:

Reviewer's Responses to Questions

**Comments to the Author**

1. If the authors have adequately addressed your comments raised in a previous round of review and you feel that this manuscript is now acceptable for publication, you may indicate that here to bypass the “Comments to the Author” section, enter your conflict of interest statement in the “Confidential to Editor” section, and submit your "Accept" recommendation.

Reviewer #2: All comments have been addressed

Reviewer #3: All comments have been addressed

2. Is the manuscript technically sound, and do the data support the conclusions?

Reviewer #2: Yes

Reviewer #3: Yes

3. Has the statistical analysis been performed appropriately and rigorously? 

Reviewer #2: Yes

Reviewer #3: Yes

4. Have the authors made all data underlying the findings in their manuscript fully available?

Reviewer #2: Yes

Reviewer #3: Yes

5. Is the manuscript presented in an intelligible fashion and written in standard English?

Reviewer #2: Yes

Reviewer #3: Yes

6. Review Comments to the Author

Reviewer #2: All the comments have been addressed.The authors have improved the quality of paper. The manuscript could be accepted.

Reviewer #3: I would like to thank to the authors for their effort of revising the manuscript. In last round I have completed my review. However, I want to point out that I have a small suggestions: I don't understand what the numbers mean on the edge of all the map figures such as fig.1-fig.6? Is that a projection coordinate values？why don't you use lat&lon? Please confirm that your map coordinate notation is correct. I have no other comments.

7. PLOS authors have the option to publish the peer review history of their article (what does this mean?). If published, this will include your full peer review and any attached files.

Reviewer #2: No

Reviewer #3: No

---

## [Editor Report · Acceptance letter]

PONE-D-24-60645R3

PLOS ONE

Dear Dr. Salmanmahiny,

I'm pleased to inform you that your manuscript has been deemed suitable for publication in PLOS ONE. Congratulations! Your manuscript is now being handed over to our production team.

Kind regards,

on behalf of

Dr. Jun Yang

Academic Editor

PLOS ONE